# Asynchronicity of deglacial permafrost thawing controlled by millennial-scale climate variability

Xinwei Yan [1,2,12], Xu Zhang [3,4,12] ✉, Bo Liu[5], Huw T. Mithan [6], John Hellstrom [7], Sophie Nuber [8], Russell Drysdale [7], Junjie Wu [9], Fangyuan Lin [10], Ning Zhao [11], Yuao Zhang[4], Wengang Kang[4] & Jianbao Liu [1,2,4] ✉

Permafrost is a potentially important source of deglacial carbon release alongside deep-sea carbon outgassing. However, limited proxies have restricted our understanding in circumarctic regions and the last deglaciation. Tibetan Plateau (TP), the Earth's largest low-latitude and alpine permafrost region, remains underexplored. Using speleothem growth phases, we reconstruct TP permafrost thawing history over the last 500,000 years, standardizing chronology to investigate Northern Hemisphere permafrost thawing patterns. We find that, unlike circumarctic permafrost, TP permafrost generally initiates thawing at the onset of deglaciations, coinciding with Weak Monsoon Intervals and sluggish Atlantic Meridional Overturning Circulation (AMOC) during Terminal Stadials. Modeling elaborates that the associated Asian monsoon weakening induces anomalous TP warming through local cloud–precipitation–soil moisture feedback. This, combined with high-latitude cooling, results in asynchronous boreal permafrost thawing. During the last deglaciation, however, anomalous AMOC variability delayed TP and advanced circumarctic permafrost thawing. Our results indicate that permafrost carbon release, influenced by millennial-scale AMOC variability, may have been a non-trivial contributor to deglacial $CO_2$ rise.

Permafrost is a fundamental component of Arctic and high-elevation landscapes, covering ~13 to 18% of the Northern Hemisphere's land surface[1]. The top three meters of permafrost are estimated to contain $1035 \pm 150$ petagrams (Pg) of soil-organic carbon, which is twice the amount currently stored in the Earth's atmosphere[2,3]. Polar and high-elevation regions have warmed faster than other areas over recent decades[2], increasing the risk of further permafrost degradation. This thawing can lead to the

[1]College of Urban and Environmental Sciences, Peking University, Beijing 100871, China. [2]Key Laboratory of Western China's Environmental Systems (Ministry of Education), College of Earth and Environmental Sciences, Lanzhou University, Lanzhou 730000, China. [3]British Antarctic Survey, Cambridge CB3 0ET, United Kingdom. [4]State Key Laboratory of Tibetan Plateau Earth System, Resources and Environment (TPESRE), Institute of Tibetan Plateau Research, Chinese Academy of Sciences, Beijing 100101, China. [5]Plateau Atmosphere and Environment Key Laboratory of Sichuan Province, School of Atmospheric Sciences, Chengdu University of Information Technology, Chengdu, China. [6]Department of Earth and Space Sciences, University of Washington, Seattle, WA, USA. [7]School of Geography, Earth and Atmospheric Sciences, University of Melbourne, Melbourne, VIC 3010, Australia. [8]School of Oceanography, University of Washington, Seattle, WA, USA. [9]Department of Environmental Science, Bolin Centre for Climate Research, Stockholm University, Stockholm 10691, Sweden. [10]State Key Laboratory of Loess and Quaternary Geology, Institute of Earth Environment, Chinese Academy of Sciences, Xi'an 710061, China. [11]State Key Laboratory of Estuarine and Coastal Research and Institute of Eco-Chongming, East China Normal University, Shanghai 200241, China. [12]These authors contributed equally: Xinwei Yan, Xu Zhang. ✉e-mail: xuang@bas.ac.uk; liujb@pku.edu.cn

release of carbon dioxide and methane from decomposing organic matter that was previously frozen[3] and creates positive feedback loops that further accelerate and amplify climate warming[4,5].

Glacial terminations (also known as deglaciations) are episodes of global climatic warming associated with increasing atmospheric carbon dioxide ($CO_2$) along with $\delta^{13}CO_2$ decrease[6–8]. The upwelling of poorly ventilated and carbon-rich abyssal Southern Ocean water masses are thought to be the primary source of deglacial $CO_2$ rise[9–11] as well as a $^{14}C$-depleted source of atmospheric radiocarbon ($\Delta^{14}C$; ref. [12]). Nonetheless, millennial-scale ventilation of Southern Ocean water masses cannot fully explain the centennial deglacial $CO_2$ and synchronous methane ($CH_4$) increases, and alone cannot account for the full magnitude of the observed decline in atmospheric $\delta^{13}CO_2$ and $\Delta^{14}C$ (refs. [4,5]). Recent evidence suggests that destabilization of permafrost carbon reservoirs could have played an important role in the last deglacial $CO_2$ rise[4,5,13–16]. However, identifying the permafrost source areas and establishing the timing of long-isolated terrestrial carbon releases during the deglaciation remains challenging.

The Tibetan Plateau (TP), with an average elevation exceeding 4000 m, is the principal area of permafrost in the low latitudes and represents the largest high-altitude permafrost region on Earth, containing a globally significant stock of approximately 160 petagrams (Pg) of soil-organic carbon[17,18]. One modeling study suggests that alpine permafrost and its associated carbon pool might be of comparable importance to that of the circum-Arctic region in determining the global permafrost-climate feedback[19]. However, the absence of detailed alpine permafrost thawing history limits our ability to obtain a comprehensive understanding of the spatial and temporal evolution characteristics, as well as the governing mechanisms of terrestrial cryosphere carbon remobilization in response to climate change. Here, we provide the orbital-scale Tibetan alpine permafrost degradation history using dated periods of local speleothem growth. We find that Tibetan Plateau permafrost primarily thawed during deglacial Terminal stadials, periods associated with a near shutdown of the Atlantic Meridional Overturning Circulation (AMOC) as manifested in millennial-scale reductions in Asian monsoon intensity (i.e., weak monsoon intervals; WMI). This contrasts with the remaining circumarctic permafrost, which typically begins thawing at the onset of interglacial periods/end of deglaciations. Given the link between permafrost thawing and the release of ancient terrestrial carbon, our data suggest that the TP permafrost could be an important contributor to the rise in atmospheric $CO_2$ during deglaciations.

Speleothems are carbonate deposits that form in karstic terrains (typically limestone and dolomite) when rainwater percolates via the soil dissolving bedrock and organic carbon. In regions where the upper vadose zone remains frozen year-round, water infiltration is halted, inhibiting the dissolution of soil-organic carbon due to reduced or absent plant activity. This, in turn, results in a cessation of speleothem growth in the underlying cave. As such, we would not expect to find speleothems in permafrost regions where the soil has remained frozen for extended periods of time. Surprisingly, ancient speleothems can be found in current permafrost zones suggesting that the overlying area may have experienced periods of thawing due to the deepening of the active layer during past warmer climates[20–22]. By reconstructing speleothem growth periods from Siberian[20,21] and Canadian caves[22,23] located in the circumarctic permafrost region, previous studies have assessed the vulnerability of circumarctic permafrost during warm interglacial periods[20–23]. In this study, we compiled 169 uranium-thorium (U-Th) dates from previous research to reconstruct the growth history of 30 speleothems from the Tianmen cave system (30°55′N, 90°4′E, ~4800 m a.s.l.) located within the discontinuous permafrost zone of the central Tibetan Plateau (Fig. 1A). In

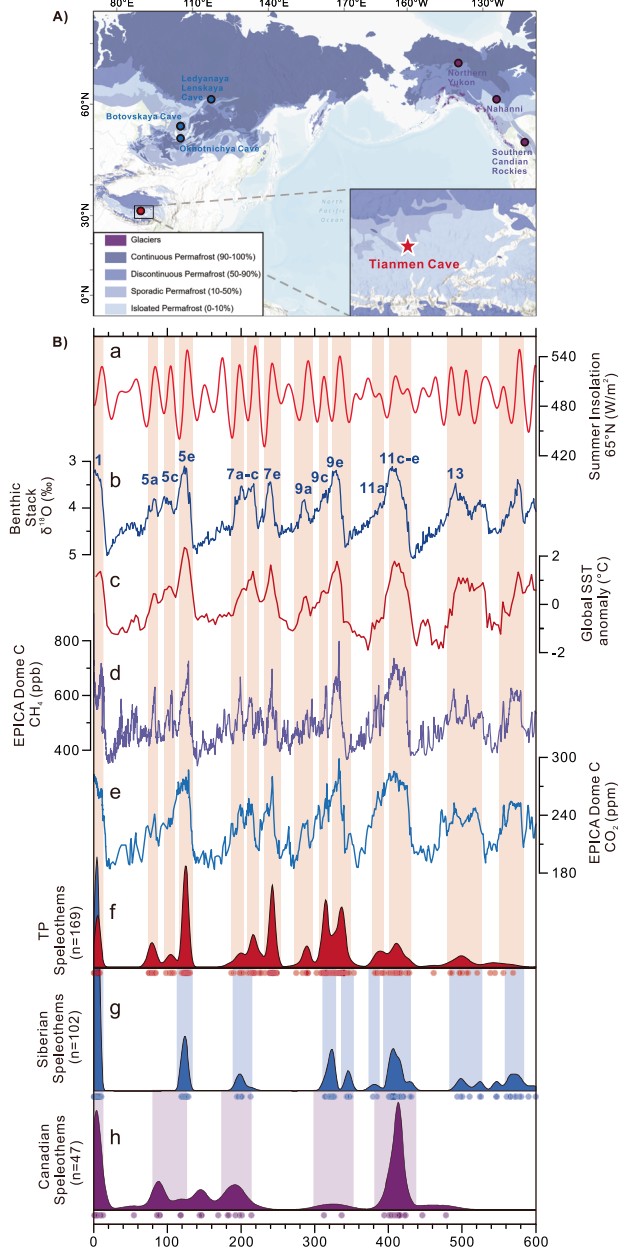

**Fig. 1 | Study area and permafrost maps, and comparison of speleothems' growth periods in Canada, Siberia, and the Tibetan Plateau with insolation and global climatic background of the past 600,000 years.** **A** Extent of permafrost types in permafrost map[75] of North America, Siberia, and Tibetan Plateau, with the location of Tibetan Plateau, Canadian, and Siberian Caves marked by red, purple, and blue circles, respectively. (This figure was created using ArcGIS® software by ESRI. Basemap sources: Esri, TomTom, Garmin, FAO, NOAA, USGS, © OpenStreetMap contributors, and the GIS User Community)[76]. **B** Red vertical bars indicate periods of growth in the Tianmen cave system, with blue and purple vertical bars representing the periods of growth in Siberian and Canadian caves, respectively. (a) Summer insolation at 65°N (ref. [77]). (b) Benthic $\delta^{18}O$ stack[78] with interglacial Marine isotope stages (MIS) numbers. (c) Standardized global sea-surface temperature (SST) stack changes[79]. (d, e) $CH_4$ and $CO_2$ records, respectively, of EPICA Dome C[80,81]. (f–h) Kernel density estimate plot showing probability densities for all published dating samples of Tibetan Plateau, Siberian, and Canadian caves, respectively, with the distribution of age determinations at the bottom. Source data are provided as a Source Data file.

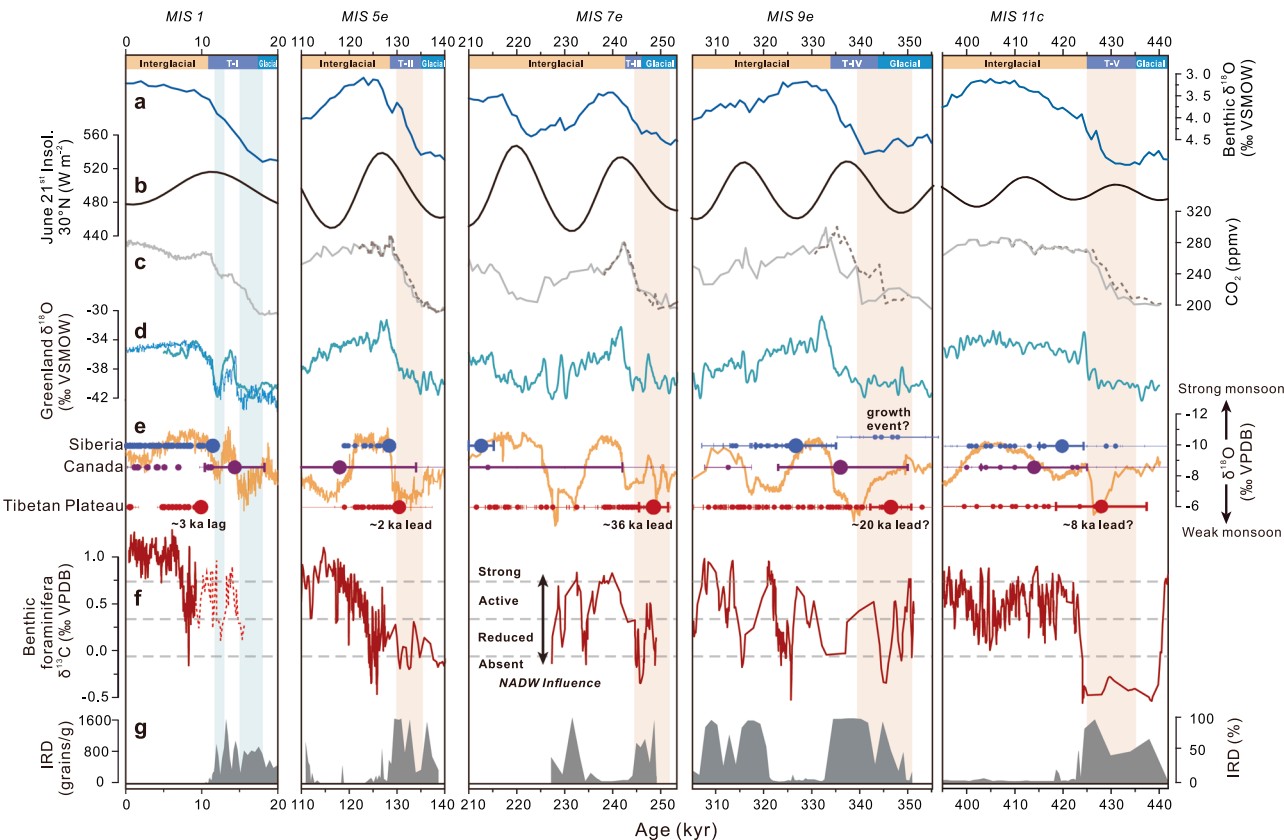

**Fig. 2 | Speleothem growth periods compared with other paleoclimate archives during interglacials Marine isotope stages (MIS) 1 to MIS 11c. a** Global benthic δ18O stack[78] as well as the CO2 concentration that has been tuned to East Asian Summer monsoon (EASM) speleothem chronology[31] (dashed dark brown). **d** Synthetic Greenland δ18O record[82] (turquoise) and NGRIP Greenland δ18O record[83] (light blue). **e** EASM strength as given by a composite Chinese cave speleothem δ18O record[31] (orange); horizontal error bars are the distribution of speleothem ages (±2σ) for Siberia (top, blue), Canada (middle, purple) and Tibetan Plateau (bottom, red), the dates of initial continuous thawing in each permafrost region are marked in bold circles (see Methods and Supplementary Fig. 2 for details). **f** Glacial-interglacial bottom water δ13C reconstructions from *Cibicidoides wuellerstorfi* in the deep Eirik Drift[37,84,85], the dashed line in MIS 1 is the bottom water δ13C reconstructions from U1304 in the North Atlantic[37], indicating the Atlantic meridional overturning circulation (AMOC) strength. **g** Eirik Drift ice-rafted debris (IRD) records[37] [percent of >150 μm entities; MIS 5e, MIS 7e, MIS 9e, MIS 11c] and ODP 983 IRD per gram[86] [MIS 1]. Red [MIS 5e to MIS 11c] and blue [MIS 1] vertical bars indicate the Terminal stadials, which correspond to the onset of Tibetan Plateau permafrost thawing except for MIS 1. Source data are provided as a Source Data file.

combination with circumarctic speleothem data, this constitutes a global-scale permafrost thawing history covering the last five glacial-interglacial cycles in the past 500,000 years (Fig. 1B and Supplementary Fig. 1). Importantly, most of the caves in these two regions are currently in the southern marginal zone of continuous permafrost and therefore sensitive to permafrost degradation in response to glacial-interglacial climate warming. This enables us to investigate the long-term tempo-spatial evolution of permafrost and the driving mechanisms, especially during terminations (Fig. 2 and Supplementary Fig. 2).

## Results and discussion
### Permafrost processes on the Tibetan Plateau
The study area on the Tibetan Plateau is characterized by a monsoonal climate with annual precipitation of about 440 mm. Around 81% of the annual rainfall precipitates in summer[24]. The mean January and July air temperatures are −12 °C and 9 °C, respectively, with mean annual air temperature (MAAT) of about −0.9 °C based on data (1955–2020) from the nearby Nagqu meteorological station (31.48°N, 92.07°E, 4508 m a.s.l.). The modern cave temperature is ~1 °C, which is ~3.7 °C higher than MAAT, assuming a lapse rate of 0.6 °C per 100 m, and the relative humidity inside the cave is ~93% (measured in August 2007)[24]. Speleothem growth in permafrost regions largely depends on a steady drip of water into the cave, which is controlled by the soil's percolation rate associated with the depth of the permafrost active layer. Rainfall

and meltwater from ground ice, the main sources of drip water[25], can seep into the cave through the overlying soil layers only when the active layer deepens due to permafrost thawing. This facilitates speleothem formation. Here, we utilize previously published U-Th dating data of speleothem calcite from active growth periods, and pair it with δ18O on the same samples to identify whether TP speleothems grew during strong or weak monsoon periods across 5 terminations during the last 500 ka. We then used kernel density estimation (KDE)[26–28] to generate probability distributions of both our compiled ages for TP[24,29,30] and published circumarctic speleothems ages from Siberia and Canada[20–23] to reconstruct active speleothem growth periods over the past ~500 kyr (Fig. 1g–i). Our findings indicate that speleothem growth in the Tianmen cave typically begins during Weak Monsoon Intervals associated with a collapsed AMOC state during climate terminations[31] (Fig. 2 and Supplementary Figs. 2, 3). These periods are characterized by drier conditions compared to interglacial climates[32]. This suggests that the overlying permafrost conditions, which influence both meltwater availability and soil percolation rates, may be the primary drivers of speleothem growth during terminations, rather than meteoric water, the modern source of drip water to the cave.

### Vulnerable Tibetan Plateau permafrost during past interglacials
The growth history of TP speleothems during the past five interglacials closely resembles that of speleothems in the circumarctic permafrost

region during these periods (Fig. 1B). Specifically, all KDE peaks and overlapping shoulders correspond consistently to typical warm interglacial periods (i.e., Marine isotope stages (MIS) 1, 5, 7, 9, 11;), indicating recurring permafrost thawing when global climate reaches a warm interglacial state, associated with high global sea-surface temperatures and greenhouse gases (GHGs). Furthermore, TP speleothems grew during interglacial sub-stages, i.e., MIS 5a, 5c, MIS 7c, 7e, and MIS 9a, while Siberian and Canadian speleothems do not show significant growth in these relatively moderate warming periods (Fig. 1B). This may be due to the higher MAAT on the Tibetan Plateau ($-4.7 \pm 4.4 \,°C$) compared to the circum-Arctic regions ($-12.4 \pm 6.4 \,°C$) resulting in a deeper active layer in the TP region[33]. This suggests that TP permafrost may be yet more vulnerable to warming climates than circumarctic permafrost, even though the latter is likely influenced by polar amplification of global warming[33] during sub-interglacial stages.

During the warm MIS 5, we observe that TP speleothem growth stagnates during cold interglacial sub-stages (i.e., MIS 5b, 5d) and during glacial inception and glaciation. These periods are characterized by local cooling associated with decreased greenhouse gas concentrations, reduced local summer insolation, and the expansion of continental ice sheets (Fig. 1B). It is likely that speleothem growth stopped due to the development of permafrost linked to regional climatic changes that inhibited water percolating into the cave. Moreover, the growth generally ends at the interglacial strong monsoon periods (Supplementary Fig. 3), minimizing the influence of rainfall-induced drainage water on speleothem growth. The discontinuities in the Holocene (MIS 1) speleothem growth for the TP may be a response to interglacial temperature fluctuations that altered the permafrost condition, and the thickness of the active layer, overlaying the specific stalagmites. For example, growth likely halted during the cooling period that followed the Holocene Climatic Optimum, and regrew during the Medieval Warm Period (Supplementary Fig. 2). This resulted in localized speleothem growth patterns within the cave, with some stalagmites growing and others remaining dormant. As an example, while the TP permafrost thawed continuously during the warm MIS 5e interglacial, individual speleothems only grew during specific periods within this stage[24]. Due to the limited number of stalagmite samples collected from the Holocene, it is possible that other speleothems in the region may have grown throughout this period but were not sampled. Nevertheless, the two existing samples support the conclusion that Tibetan Plateau stalagmites did not begin to grow until the onset of the Holocene.

**Onset of Tibetan permafrost thawing during deglacial Terminal stadials**

The U-Th chronology underpinning the speleothem growth phases constrains the initial date of continuous permafrost thawing (Methods) during the past five glacial-interglacial cycles. During terminations (T)-II and T-III, the onset of circumarctic permafrost thawing is confined to interglacial periods, and lags TP permafrost thawing (Fig. 2). During T-IV, Siberian speleothems exhibit an unusual growth phase preceding MIS 9e. This is identified through four ages from four individual speleothems, and is better characterized as a "growth event" rather than a phase of continuous growth. The cause of this glacial growth event is undetermined. Glacial low biogenic silica content in nearby Lake Baikal suggests it is unlikely to have been driven by climate warming (Supplementary Fig. 4). Consequently, this glacial growth event may not be directly linked to continuous permafrost thawing suggesting that the thawing of TP permafrost likely preceded that of the circumarctic region. Given the complexity of thawing mechanisms and the scarcity of regional temperature records, further research is necessary to fully understand the causes of this early growth event. During T-V, the large dating uncertainties in circumarctic samples make it challenging to establish the lead-lag relationship between TP and circumarctic permafrost thawing.

Nevertheless, when considering the dates with relatively lower age uncertainties as the onset of continuous speleothem growth (Methods), it is possible that Tibetan Plateau permafrost thawing may still have preceded that of the circumarctic region. Overall, despite the complexities in the timing of circumarctic permafrost thawing, the onset of Tibetan Plateau permafrost thawing generally occurs within dating uncertainties during deglaciations, prior to the continuous thawing of circumarctic permafrost during warm interglacial periods/ end of the deglaciation (except for T-I) (Fig. 2 and Supplementary Fig. 2).

Notably, the onset of TP permafrost thawing coinciding with the WMI and the collapse of the AMOC[31,34] appears to be a common feature. This is supported by higher $\delta^{18}O_{carbonate}$ values in both the East Asian summer monsoon (EASM) composite speleothem record[31] and Tianmen cave records which serve as proxies for the Southern Asian summer monsoon (SASM)[24] (Fig. 2 and Supplementary Fig. 3). The $\delta^{18}O$ values of speleothems in the Asian summer monsoon (ASM) domain are primarily controlled by summer monsoon intensity which is driven by large-scale changes in atmospheric circulation accompanied by changes in rainfall amount, temperature, seasonality, as well as moisture source and trajectories (ref. 24 and references therein). It has been suggested that changes in ASM circulation manifest as millennial-scale positive $\delta^{18}O$ speleothem anomalies, are a consequence of AMOC slow down during Terminal stadials[31,34]. This correlation further indicates that Tibetan permafrost thawing and deglacial AMOC reductions are likely in close association, independent of the absolute speleothem dates. Moreover, the $\delta^{18}O_{carbonate}$ record of a central European Cave[35] and $\delta^{13}C_{carbonate}$ record from a cave fronting the North Atlantic[36] provide good constraints on the timing of AMOC disruption and the last warming stage of the T-II (Supplementary Fig. 5). By comparing our onset dates of continuous growth with these absolute dated record from the high-latitude regions, we find that TP permafrost thawing occurred during AMOC collapse of T-II, while the circumarctic permafrost thawed during the following warm interglacial period (MIS 5e). This well-constrained termination corroborates AMOC-induced TP permafrost thawing inferred from ASM $\delta^{18}O$ records.

While the chronology of marine records is less precise than that of absolutely dated speleothems, various lines of marine evidence consistently demonstrate that glacial terminations are characterized by a sequence of prominent climatic events. These include increased iceberg rafting (Fig. 2g) and depleted benthic foraminifera $\delta^{13}C$ values in the North Atlantic (Fig. 2f), indicating a shallower or weaker AMOC at the onset of the termination[37], which precedes the main phase of deglaciation as manifested in benthic $\delta^{18}O$ records (Fig. 2). This has been widely interpreted as early deglacial freshwater input disrupting AMOC[37]. These "Heinrich Events" are well reflected in absolutely dated ASM speleothem records and support the link between continuous TP permafrost thawing and early deglacial AMOC disruptions.

Furthermore, the initial growth rate of Tibetan Plateau speleothems was generally higher at the onset of deglaciations compared to subsequent interglacial periods (Supplementary Fig. 6), which cannot be explained by the drier climate during terminations. Instead, this high growth rate suggests abrupt thawing of ground ice and increased percolation rate, rather than direct atmospheric precipitation supply, as the primary controls for speleothem growth. Additionally, permafrost thawing can enhance the respiration of previously frozen soil carbon as evidenced by the decreased $\delta^{13}C$ values of stalagmite at the onset of thawing[38] (Supplementary Fig. 6). The release of this freeze-locked carbon can elevate $pCO_2$ levels in the soil potentially leading to higher $Ca^{2+}$ concentrations in drip waters and stimulating faster speleothem growth[38].

It is also noteworthy that the last deglaciation was an exception. Unlike the circumarctic permafrost, which thaws in the middle of the deglaciation, Tibetan Plateau permafrost did not thaw until the early Holocene (Fig. 2). This suggests that increasing GHGs and summer

insolation during termination are not the dominant factors controlling TP permafrost thawing. Instead, we attribute this exception to the early recovery of the AMOC during Bølling-Allerød, coupled with an active AMOC during Heinrich Stadial 1 (HS1), which effectively reduces TP warming during HS1 and postpones the onset of TP permafrost thawing during T-I.

Given the varying characteristics of the past five terminations, we acknowledge the complex relationship of the TP permafrost thawing with changes in summer insolation, ice volume, and GHGs. For example, the onset of continuous thawing does not align with changes in local summer insolation during deglaciations, since TP permafrost thawing occurs during both high (T-II and T-III) or low (T-IV and T-V) rates and magnitudes of summer insolation increase (Fig. 2b). Furthermore, there is no clear relationship between permafrost thawing and GHG levels. During T-II and T-V, speleothem growth begins during the final stages of deglacial $CO_2$ rise, while during T-III and T-IV, speleothem growth commences during the initial stages (Fig. 2c). This aligns with the thawing dynamics of TP permafrost, which can occur across a wide range of ice volumes and $CO_2$ levels (Fig. 2a, c). Consequently, the lead-lag relationships between the Tibetan Plateau and circumarctic permafrost thawing may vary among different terminations. To clarify these differences, more precisely dated speleothems from permafrost regions covering T-IV and earlier periods are needed.

## Dynamics of boreal permafrost thawing during deglaciations

The AMOC plays a fundamental role in hemispheric heat distribution by regulating meridional heat transport (Supplementary Fig. 7). This regulation is driven by changes in the formation and export of North Atlantic Deep Water[37,39]. Multiple climate models suggest that a collapsed AMOC could lead to warming in northern India and the TP[40–43], although the underlying mechanism remains to be discussed. The warming pattern contrasts with the cooling observed in other regions at similar latitudes to the Tibetan Plateau and northern India, such as the Mediterranean region[44] (Fig. 3). This suggests that the warming in these regions cannot be solely attributed to meridional heat redistribution caused by the AMOC collapse. A comprehensive analysis of the factors affecting the surface heat balance in the TP region is necessary to understand the underlying dynamics driving its warming. Here we use a climate feedback-response analysis method (CFRAM) to quantify the contributions of different physical and dynamical processes (Fig. 3).

Our analysis suggests that a collapsed AMOC could shift the Intertropical Convergence Zone (ITCZ) southward due to meridional heat redistribution[45] leading to a weakening of the ASM (Fig. 2e). This, in turn, could reduce moisture transport from the ocean resulting in decreased cloud cover and precipitation in northern India and the Tibetan Plateau. Studies have shown that the abundance of mixed-phase clouds in summer can have a significant cooling effect on the TP[46]. Reduced cloud cover can lead to increased downward shortwave radiation resulting in a positive cloud radiative effect at the surface (Fig. 3d). Consequently, reduced rainfall can dry the ground, reducing soil moisture and associated surface evaporation (latent heat flux) and further contributing to warming (Fig. 3f). Moreover, the reduced soil heat capacity of drier ground can also contribute to surface warming (Fig. 3h). Collectively, a collapsed AMOC could significantly contribute to both summer and annual warming on the TP (Fig. 3a and Supplementary Fig. 8) through local cloud−precipitation−soil moisture feedbacks. Using a permafrost heat diffusion model (Methods), we confirm that the AMOC-induced TP warming during Terminal stadials can effectively deepen the permafrost active layer by promoting its thawing (Supplementary Fig. 9). As a result, associated meltwater as well as precipitation would penetrate through the cave causing speleothem growth.

With the collapse of the AMOC, northward heat transport was significantly reduced, leading to a pronounced cooling in the northern high latitudes (Fig. 3). This initial cooling was evident in the North Atlantic and accompanied by sea-ice expansion which subsequently influenced the North American and Eurasian continents through the ice-albedo feedback[47]. These changes effectively offset the effects of rising $CO_2$ and summer insolation on local surface temperatures delaying circumarctic permafrost thawing until the onset of interglacials/end of deglaciations. At this time, the AMOC recovers to its active mode coinciding with high $CO_2$ levels and low ice volumes/extents. This period is characterized by a generally ameliorated northern high-latitude climate (Fig. 2d).

A notable finding is that Tibetan Plateau permafrost did not thaw until the Holocene, while circumarctic permafrost experienced an earlier thawing during the last deglaciation (Fig. 2). This is consistent with previous circumarctic permafrost studies of marine sediments[15,16] and model simulations[4,5] suggesting remobilization and release of circumarctic permafrost carbon during the last deglaciation. Compared to previous terminations, the Bølling-Allerød event during T-I (ref. 48) effectively mitigated the warming effect of TP due to the rapid recovery of the AMOC during the middle stage of the deglaciation. In contrast, this abrupt AMOC recovery triggered the thawing of circumarctic permafrost.

The onset of TP permafrost thawing generally occurred during the middle or end of the WMI during T-II to T-V (Fig. 2). This suggests that the brief early disruption of the AMOC during HS1 was unable to deepen the active layer and initiate speleothem growth due to insufficient heat accumulation. Furthermore, there is increasing evidence suggesting that the AMOC remained relatively active during HS1 (refs. 49–51) and the Younger Dryas[52], despite being weaker than its state during the Last Glacial Maximum[50,53]. We, therefore, suggest that the weaker but still active AMOC during HS1 cannot give rise to robust warming and hence delay the TP permafrost thawing during T-I. Despite the lack of comparative studies on the AMOC activity during different terminations, it has been demonstrated that a weakened AMOC can produce anomalous warming in the Indo-Pacific ocean[54,55] (Supplementary Fig. 10). Records suggest that Indian Ocean Sea-surface temperature (SST) warming relative to the North Atlantic was less pronounced during T-I compared to T-II (Supplementary Fig. 11). This may suggest that the AMOC was more active during T-I than T-II. Additionally, carbon isotope ratios of benthic foraminifera in the North Atlantic (ODP Site 983) likely suggest a stronger AMOC during T-I compared to the previous four terminations (Supplementary Fig. 12). These observations strengthen the critical role of AMOC changes in determining the onset of permafrost thawing during deglaciations. Therefore, the observed phase difference in the timing of permafrost thawing between the circumarctic and TP regions can be explained by changes in the strength of the AMOC that exerts opposing thermal effects in the two regions (Fig. 4).

There is no TP permafrost thawing events are observed in our records during MIS 3, despite repeatedly occurring Heinrich events. This might be related to the shorter duration of these events and the more stable, cold glacial climate background compared to terminations (Supplementary Fig. 10), which effectively reduces the AMOC-induced TP warming that leads to permafrost thawing. The lower temperature during MIS 3 may have resulted in colder underlying soil and bedrock, making the region less susceptible to AMOC-induced warming in the TP. Further research using permafrost proxies from other regions of the Tibetan Plateau is essential for understanding the spatial and temporal characteristics, as well as the underlying dynamics, of permafrost changes of the TP across glacial-interglacial cycles.

Our findings suggest that permafrost thawing is a common occurrence during the past five deglaciations. Tibetan alpine

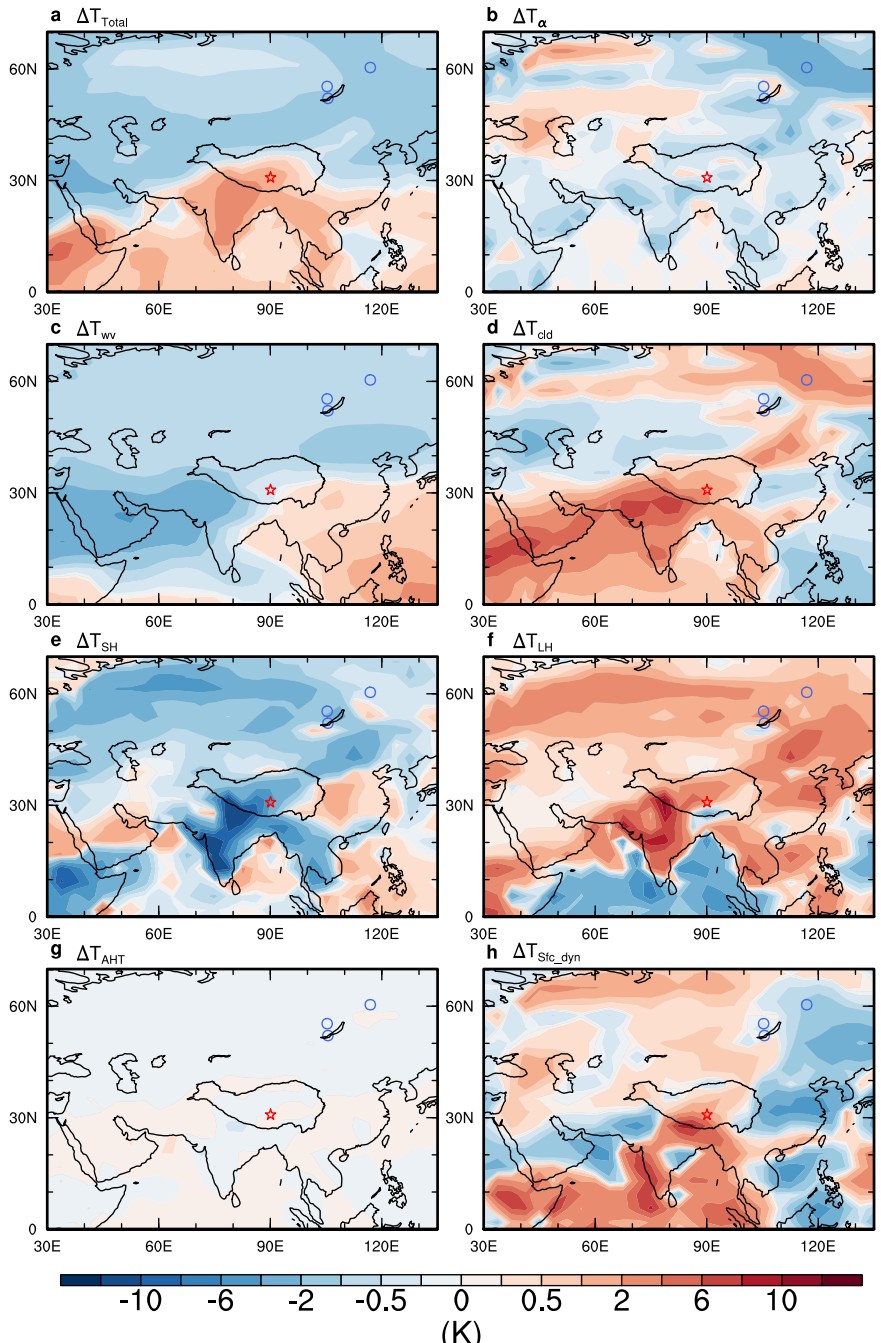

**Fig. 3 | Processes contributing to Tibetan Plateau warming in boreal summer.** **a**–**h** Decomposition of **a** surface temperature difference between the North Atlantic hosing experiment (GFWF) and the equilibrated glacial control simulation (GCTL) into seven individual processes due to **b** surface albedo, **c** water vapor, **d** cloud, **e** surface sensible heat flux, **f** surface latent heat flux, **g** atmospheric heat transport, and **h** surface dynamics (i.e., land and oceanic surface energy transport). The Tibetan Plateau caves and the Siberian Caves are marked by red stars and blue circles, respectively. (Map produced by NCL Version 6.6.2, https://doi.org/10.5065/D6WD3XH5)[87].

permafrost thawing occurred during the Terminal stadials, which preceded the circumarctic permafrost thawing in most terminations. This stands in contrast to the circumarctic regions where permafrost thawing is evident at the end of the deglaciations. Modeling results further reveal that the asynchronous thawing between the Tibetan Plateau and circumarctic permafrost is controlled by the magnitude of millennial-scale climate variability during deglaciations. The last deglaciation appears unusual in comparison to other terminations, due to the advanced thawing of the Tibetan Plateau permafrost and delayed response in the circumarctic regions. Conclusions, therefore, need to be made separately for T-I and the other Termination

events. Nonetheless, thawing permafrost, irrespective of location, always presents a carbon source for deglacial atmospheric $CO_2$ rise and, therefore, has the potential to close the glacial-interglacial cycle carbon budget when considered in addition to the marine carbon pool[4,5,15]. In addition, our findings might shed light on mechanisms of centennial-scale $CO_2$ pulses during deglaciations and warm interglacials[56,57]. Current Earth System Models[3] tend to parameterize climate-permafrost-carbon feedbacks. It may be beneficial to include them in future model set-ups to advance our understanding of how permafrost carbon may have driven and will drive past, current, and future climate variability.

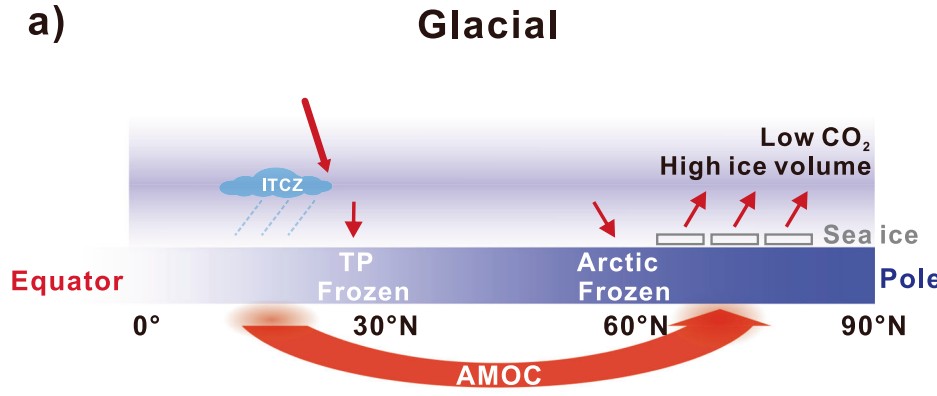

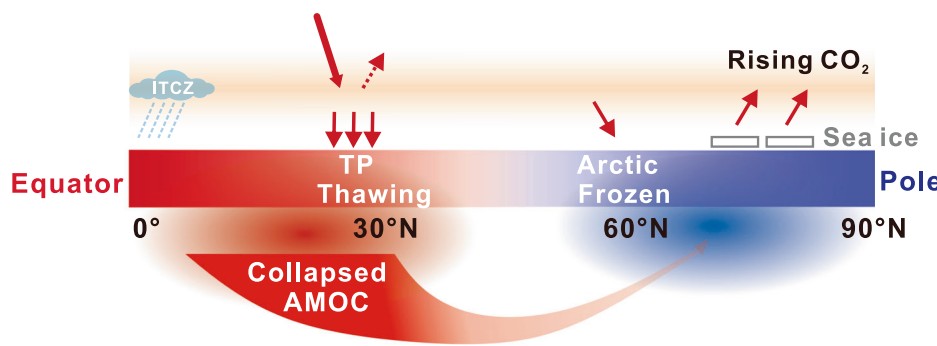

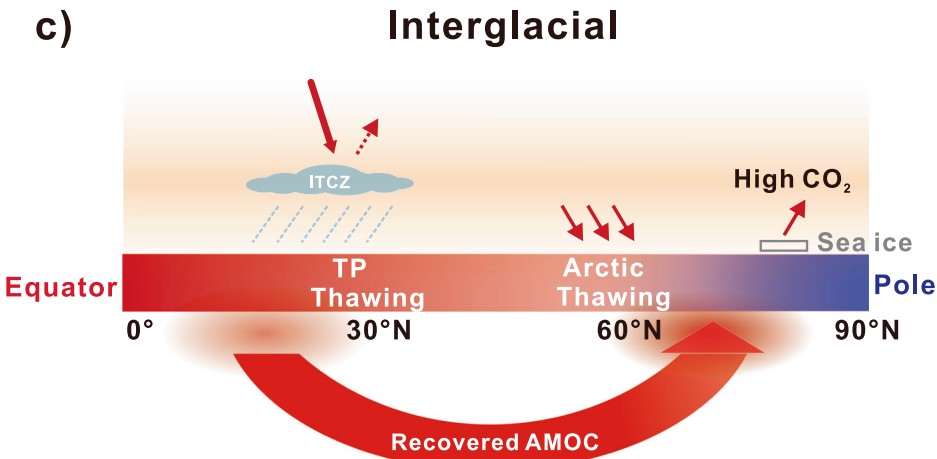

**Fig. 4 | Conceptual diagram showing the relationship between Atlantic meridional overturning circulation (AMOC) variations and permafrost thawing during glacial-interglacial cycles. a** Frozen permafrost in Northern Hemisphere during the cold glacial period with low greenhouse gases (GHGs). **b** The deglacial collapse of the AMOC reduces polar energy transport, leading to cooling in the circumarctic region. Concurrent reductions in South Asian summer monsoon induce Tibetan Plateau (TP) warming through local cloud–precipitation–soil moisture feedback. This, along with rising greenhouse gas levels, contributes to TP permafrost thawing. **c** The recovery of AMOC during interglacial periods enhances polar energy transport and, in conjunction with elevated GHGs, contributes to warming in high-latitude regions and the thawing of circumarctic permafrost.

## Methods
### Study area
Tianmen Cave system (30°55′N, 90°4′E, ~4800 m a.s.l.) is located between the Tanggula and Nyainqentanglha Mountains, in Bange County, Nagqu City, south-central Tibetan Plateau, China (Fig. 1A). The cave is located on the plateau few tens of meters above a river. The absence of stream channels above the cave prevents the formation of "Taliks" (unfrozen patches in the permafrost beneath a stream

channel) through which the water can infiltrate down into the cave. The cave is less than 50-m deep and extends relatively horizontally. The epikarst above the cave is approximately 10 meters thick on average. The relative humidity within the cave is ~93%, whilst the temperature is ~1 °C (measured in August 2007), which is ~2 °C higher than the local mean annual temperature (−0.9 °C) in Nagqu meteorological station (31.48°N, 92.07°E, 4508 m a.s.l., ~190 km to the east of Tianmen Cave). The mean annual precipitation is 442 mm based on Nagqu station data from 1955 to 2020 CE; over 81% of local precipitation falls in summer months (June-September), when the South Asian Summer Monsoon prevails[24]. The vegetation cover comprises an alpine meadow. The cave entrance is very small (~2 m × 1 m), which is favorable to maintain a constant temperature inside the cave.

### Speleothem age data compilation

Our compiled speleothem age data were obtained from the published literature[20–24,29,30], comprising 377 uranium-thorium and 10 uranium-lead dates of speleothem growth in Arctic and Tibetan permafrost regions whose ages span ~600 ka to the present (n = 308). All ages were recalculated using the most recent estimates of the decay constants of $^{234}U$ ($\lambda_{234} = 2.82206 \times 10^{-6}$) and $^{230}Th$ ($\lambda_{230} = 9.1705 \times 10^{-6}$) (ref. 45). Detrital thorium correction is essential to U-Th dating and can be performed by estimation of the initial $^{232}Th/^{230}Th$ activity ratio using one of several potential methods[58]. The $^{232}Th/^{230}Th$ activity ratio used in the original publication was preserved when based on isochron techniques or stratigraphic constraints specific to that setting. By far, the most common approach assumes a bulk Earth $^{232}Th/^{230}Th$ activity ratio of $0.82 \pm 0.41$ (equivalent to an atomic ratio of $4.4 \pm 2.2 \times 10^{-6}$), derived from the bulk Earth $^{232}Th/^{238}U$ atomic ratio of $3.8 \times 10^{-6}$, but this does not encompass the full range of $^{230}Th/^{232}Th$ variability observed in speleothems[58]. Where the bulk Earth value had previously been assumed, it was replaced by a $^{232}Th/^{230}Th$ activity ratio of $1.5 \pm 1.5$ which has been shown to be a more appropriate estimate where no prior information is available, given the range of values seen in speleothems[58]. Where necessary, stratigraphic constraints[58] were used to refine this to a speleothem-specific estimate. In all other cases, where it was otherwise unclear what (or if any) correction had been performed, a $^{232}Th/^{230}Th$ activity ratio of $1.5 \pm 1.5$ (2σ) was applied. Shifting to this correction value resulted in a slightly larger correction leading to slightly younger corrected ages for samples with a significant detrital component. The uncertainty associated with correction for detrital thorium was fully propagated to the final age. All recalculated ages are expressed as "thousands of years before present", where "present" is the year 1950 CE. U-Th ages are calculated in years before the day on which the U-Th measurements were made. In cases where the year of measurement was not given in the literature, it was assumed to be 2 years prior to the date of publication of the record. This was performed for consistency and makes an infinitesimal difference to the final calculated ages or any comparisons. The difference between the recalculated results and the originally published dates is less than 2%; therefore, using either dataset does not affect the conclusions.

In short, there are 102 dates of 21 individual speleothems from East Siberia and 47 dates (including 22 that used low-precision alpha spectrometry dating methods) of 30 individual speleothems from West Canadian permafrost regions and Bear River Range-Uinta Region, USA (2 dates). The low-precision alpha spectrometry[22] data is less constrained on probability density peaks; for example, the anomalous glacial peak during MIS 6 (Fig. 1B) is probably due to the large dating uncertainties (Supplementary Fig. 1), or these samples were collected from cave entrances, which are subjected to seasonal melting and cannot be attributed to deep ground thaw that we discuss here. Here we focus on the well-constrained Siberian permafrost data as well as the low-latitude speleothem growth in the Tibetan permafrost region. The Tibetan data is collected from all published uranium-thorium ages

(n = 169) of 30 speleothems derived from two caves within this region. Most speleothem samples (n = 27) are from Tianmen Cave. Three speleothems were recovered from nearby Dimen Cave (~300 m to the west). The number of samples included in this dataset is greater than the reasonable size for reliably differentiating between random versus non-random events[26]. To reconstruct a composite and, as continuous as possible, Indian summer monsoon hydroclimate history using speleothem oxygen isotopes, the speleothems were retrieved where available throughout the caves[24]. The dating samples included in this dataset have been collected for the specific purpose of constraining changes in deposition modes and initial/terminal growth histories of the speleothems, hence the dataset represents an essentially random sample and is well-suited for examining permafrost degradation trends. It should be pointed out that the sub-samples for U-Th analyses in Tibetan speleothems were taken without deliberate selection of the growth boundaries, but very close to them[24]. While most dating in circumarctic samples were taken as close as possible to growth hiatuses (sampling at a hiatus risks incorporating a small amount of "contaminating" material from the other side of the hiatus, therefore samples were taken a small distance from hiatuses)[21,23]. This will lead to even slightly earlier thawing of TP permafrost than the dates presented. This limitation needs further work to improve, but does not change our conclusions. Moreover, the earlier deglacial growth of TP speleothems are an unambiguous phenomenon that occurred during all investigated terminations except T-I (i.e., T-II to T-V), indicating that this early growth is non-random and unaffected by artificial sampling biases.

Speleothem growth frequency was identified by a kernel density estimator (KDE)[26], which is more accurate than probability density plots, as peaks in the latter often reflect data precision rather than data density. In this approach, periodic permafrost dynamics were recovered from growth-frequency distributions derived from radiometric ages. The KDE distribution was constructed with DensityPlotter software[59] using adaptive bandwidths of 3.98, 2.87, and 7.54 ka (determined using the approach detailed in Silverman (2018))[60] for Tibetan, Siberian, and Canadian speleothem ages, respectively.

### Criteria for continuous growth and selection of the dates representing the onset of continuous thawing

For slow-growing speleothem, verifying if the growth is continuous is difficult. In this study, the "continuous growth" phase is defined as an increment of speleothem deposition supported by multiple consecutive ages without evidence for visual discontinuity of growth, and with the primary growth period occurring during warm interglacial periods, as suggested by the established causality between speleothem growth and overlying ground thawing (Supplementary Fig. 2).

The dates with overly large uncertainties (e.g., spanning whole interglacial or glacial periods) would undermine other valuable and reliably finely dated samples, making them worthless for providing useful information when investigating the initial timing of growth. It is reasonable that the dates with lower uncertainty can provide more valuable information. Therefore, the dates with relatively lower dating uncertainties were assumed as the most reliable for estimating the onset timing for continuous permafrost thawing.

### Temporal diversity of deglacial permafrost thawing

Given that the climatic background (global ice volume, greenhouse gas level, insolation) is different for each termination and interglacial, it is reasonable to have different lead-lag features between TP and circumarctic permafrost thawing among each termination. For instance, during T-II, the lag of Siberian permafrost thawing (~128 ka) relative to TP (~130 ka) during T-II is at least ~2 ka, and the lag of Canadian permafrost thawing is even larger. During T-III, the onset of Siberian

thawing (-213 ka) lags the TP case (-249 ka) by about 36 kyr. This large time lag is a consequence of the circumarctic speleothem growth occurring at MIS 7a-c by skipping MIS 7e. The average atmospheric $CO_2$ level during MIS 7e is lower than that of MIS 5e, MIS 9e, and MIS 11c, suggesting a lower climatology air temperature during this interglacial sub-stage with a very short duration, corroborated by lower biogenic silica concentration in nearby Lake Baikal (Supplementary Fig. 4). Therefore, discontinuous/sporadic permafrost and shallower active layer may dominate circumarctic permafrost cave systems, and the chances for growing speleothem under permafrost is low during MIS 7e. During MIS 7a-c, the relatively warmer climate, as indicated by ice core $CO_2$ records, may have led to persistent thawing circumarctic permafrost. In contrast, the generally warm background in low latitudes resulted in a deeper active layer in TP, which increases the sensitivity of TP permafrost to warming, i.e. the TP permafrost region requires a shorter warming duration to deepen the active layer, which effectively enables thawing water to penetrate vadose zones to the cave system. This accounts for the persistent speleothem growth throughout MIS 7a-e. During T-IV, the lag of Siberian (-327 ka) relative to TP permafrost thawing (-347 ka) is about 20 ka after ruling out a preceded temporary growth event in Siberia (Methods; Supplementary Fig. 2). The greater error of the older U-Th dates making it difficult to make similar comparisons for T-V. Nonetheless, it is reasonable that the dates with lower uncertainties can provide more valuable information. Therefore, the dates having relatively low dating uncertainties were assumed as the most reliable for estimating the onset timing for continuous permafrost thawing (Methods). Based on this criterion, we suggest that the lag of circumarctic permafrost thawing to that in the TP is larger than -8 kyr, quantitatively consistent with T-II and T-IV (Supplementary Fig. 2).

## Climate model and experimental details

In this study, we used the comprehensive, fully coupled atmosphere–ocean general circulation model (AOGCM), COSMOS (ECHAM5-JSBACH-MPI-OM). The atmospheric model ECHAM5[61], complemented by the land surface component JSBACH[62], is used at T31 resolution (~3.75°), with 19 vertical layers. The ocean model MPI-OM[63], including sea-ice dynamics that is formulated using viscous-plastic rheology[64], has a resolution of GR30 (3° × 1.8°) in the horizontal, with 40 uneven vertical layers. The climate model has already been used to investigate a range of paleoclimate phenomena[65–69], especially millennial-scale abrupt glacial climate changes[43,67,70,71]. This indicates that it is capable of capturing key climatic features associated with abrupt shifts in the operating mode of the AMOC and is thus a very suitable climate model for this study.

To test the impacts of an "off" mode of the AMOC on the Tibetan Plateau during deglaciations, we first conducted an equilibrated glacial control simulation (GCTL) by imposing the fixed boundary conditions of the Last Glacial Maximum as proposed by the PMIP3 protocol. GCTL is integrated for 3000 years to an equilibrated state that serves as a basis for a classic North Atlantic (NA) hosing experiment (GFWF) in which a freshwater flux of 0.2 Sv ($1\ Sv = 10^6\ m^3/s$) was imposed into the Ice-Rafted Debris Belt for 800 years to shut down deep-water formation in the North Atlantic. In this study, GFWF is not aimed at reproducing HS1 as in previous studies but rather mimicking a generic deglacial Heinrich event with an "off" mode of the AMOC. The average of the last 100 years in GFWF is calculated to represent the climatology.

## Climate feedback-response analysis method

In the framework of the climate feedback-response analysis method (CFRAM)[72,73], the temperature difference between two long-term mean states can be decomposed into several partial components due to various physical and dynamical processes, and the sum of these components is equal to the total temperature difference. In this study, we selected the last 100 years of both the equilibrated glacial control

simulation (GCTL) and the North Atlantic hosing experiment (GFWF) as the equilibrium states, and decomposed the temperature differences between the two mean states using CFRAM. The formula of CFRAM used here is as follows:

$$\Delta\vec{\mathbf{T}} = \left(\frac{\partial\vec{\mathbf{R}}}{\partial\vec{\mathbf{T}}}\right)^{-1}\left(\Delta\vec{\mathbf{S}}^{\alpha} + \Delta\left(\vec{\mathbf{S}} - \vec{\mathbf{R}}\right)^{wv} + \Delta\left(\vec{\mathbf{S}} - \vec{\mathbf{R}}\right)^{cld} + \Delta\vec{\mathbf{Q}}^{SH} + \Delta\vec{\mathbf{Q}}^{LH} + \Delta\vec{\mathbf{Q}}^{AHT} + \Delta\vec{\mathbf{Q}}^{Sfc_{dyn}}\right)$$

(1)

where $\vec{\mathbf{T}} = \left(T_1, T_2, \dots, T_{Surface}\right)$ is the temperature vector from surface to the top level of the atmosphere, $\vec{\mathbf{S}} = \left(S_1, S_2, \dots, S_{Surface}\right)$ is the convergence of net shortwave radiation fluxes, $\vec{\mathbf{R}} = \left(R_1, R_2, \dots, R_{Surface}\right)$ is the divergence of net longwave radiation fluxes. The term $\frac{\partial\vec{\mathbf{R}}}{\partial\vec{\mathbf{T}}}$ is the Planck feedback matrix. The superscripts $\alpha$, $wv$, and $cld$, represent surface albedo, atmospheric water vapor, and cloud, respectively. $\Delta\vec{\mathbf{Q}}^{SH} = (0, 0, \dots, 0, -\Delta SH)$ and $\Delta\vec{\mathbf{Q}}^{LH} = (0, 0, \dots, 0, -\Delta LH)$ denote energy perturbations due to changes in surface sensible and latent heat fluxes, respectively. Note that surface sensible (SH) and latent (LH) heat fluxes are upward positive. $\Delta\vec{\mathbf{Q}}^{AHT}$ and $\Delta\vec{\mathbf{Q}}^{Sfc_{dyn}}$ represent perturbations of the energy convergence due to atmospheric heat transport (AHT) and surface dynamics, respectively. By definition, $\Delta\vec{\mathbf{Q}}^{AHT}$ has zero values at the surface layer, and $\Delta\vec{\mathbf{Q}}^{Sfc_{dyn}}$ has zero values in all atmospheric layers.

In summary, the partial temperature differences ($\Delta\vec{\mathbf{T}}^{(x)}$) due to the above-mentioned seven individual processes can be expressed as:

$$\Delta\vec{\mathbf{T}}^{(x)} = \left(\frac{\partial\vec{\mathbf{R}}}{\partial\vec{\mathbf{T}}}\right)^{-1}\Delta\vec{\mathbf{F}}^{(x)}$$

(2)

where $\Delta\vec{\mathbf{F}}^{(x)}$ denotes energy perturbations due to the seven processes, namely, differences in surface albedo, water vapor, cloud, surface sensible heat flux, surface latent heat flux, atmospheric heat transport, and surface dynamics from Eq. (1). With the aid of Eqs. (1–2), we can manage to quantify the contributions of these seven processes to the temperature difference between GFWF and GCTL.

## Permafrost heat diffusion model

We used the widely published and applied permafrost heat diffusion model as defined in ref. 1 and in the equation below to describe expected temperature diffusion patterns into the permafrost:

$$T_z = T_a + A_s * e^{-z\sqrt{\frac{\pi}{K_a * p}}}$$

(3)

Where $T_z$ is the resulting temperature at depth $z$, $T_a$ is the annual mean surface temperature, $A_s$ is the annual mean surface temperature amplitude defined as annual max temperature + annual mean temperature, $K_a$ is the diffusivity of the ground above the cave, and $p$ is the period of time reflecting the temperature parameters (here annual, 365 days). The resulting plots show the depth of the permafrost active layer, where $T_z > 0\ °C$ (Supplementary Fig. 9). The active layer defines the depth to which the ground thaws and re-freezes on an annual basis, while the underlying ground stays frozen (permafrost). As such, the underlying permafrost can only thaw due to significant and prolonged climate warming.

As evident from the equation above (3), the permafrost diffusion curves are highly dependent on the temperature information that feeds into the equation. We therefore collected annual temperature data from our model runs (Supplementary Table 1; Supplementary Fig. 13), and compared the temperature output from the Holocene

"piCTRL" to published temperature information from the Tibetan Plateau[74].

Our modeled Holocene annual temperature amplitude of −13.03–7.90 °C, and annual average of 0.69 °C, for the Tibetan Plateau are in good agreement with the annual amplitude of −12 to 8 °C, and annual averages across sites of −1 to −4 °C reported by ref. 74. After choosing a representative thermal diffusivity ($K_a$) of 0.02 mm²/s, we receive a comparable trumpet temperature curve to the measured curve by ref. 74, with a similar Holocene active layer depth of 5 m.

Next, we identified the average annual surface temperature and average annual surface temperature amplitude from the model runs for "LGM + AMOC off (hosing)", and "LGM + AMOC on (CTRL)", and used them to calculate the diffusion curves for those climates. As expected, the permafrost was, in general, colder and more stable during the glacial maximum compared to the Holocene, and the active layer was shallower (Supplementary Fig. 9). Comparing the "hosing" experiment with the "CTRL" experiment highlights that the different AMOC conditions cause a small shift in the temperature regimes leading to a slightly higher mean annual surface temperature, and a greater mean annual surface temperature amplitude in the "hosing" compared to the "CTRL" experiment. The equation above, as well as the trumpet curves, show that such a shift in the temperature regime will inevitably lead to a deepening of the active layer and a thawing of the adjacent permafrost. Based on field observation, water dripping is very active in the cave. This suggests that the cave ceiling is currently located within the active layer (i.e., shallower than 5 m), rather than permafrost. Our ages from the Last Glacial Maximum suggest no growth prior to changes in the AMOC. We take this as evidence that the cave ceiling must have been located deeper than 2 m, which is our modeled active layer depth for the LGM. The regional temperature shift after AMOC change (i.e., hosing vs CTRL experiments) would be sufficient to deepen the active layer by 30 cm on the Tibetan Plateau, which is substantial in terms of thawed ground volume, and which must have impacted the cave system. In combination with our age data, we suggest that this has a significant impact on the amount of water entering the cave and commencing speleothem growth.

## Data availability
All data were available in the manuscript and the supplementary materials. Source data are provided with this paper.

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

## Acknowledgements
We thank Anton Vaks, Haibo Wang, Yanjun Cai, Nicole Biller-Celander, Russell Harmon, and their co-authors for the exhaustive studies of cave speleothems in the permafrost regions. This study is supported by Natural Science Foundation of China (42225105), the National Key Research and Development Projects of China (2023YFF0805201), Basic Frontier Science Research Program From 0 to 1 Original Innovation Project of CAS (ZDBS-LY-DQC006), and the Innovation Program for Young Scholars of TPESER (TPESER-QNCX2022ZD-02).

## Author contributions
X.Y., X.Z. and J.L. conceived the idea. X.Y. collected and interpreted speleothem records and collated and standardized the records with the aid of J.H. and R.D. X.Z. performed climate model simulations and analyzed model outputs with the aid of B.L. and Y.Z. H.T.M. and S.N. conducted permafrost modeling. X.Z. and X.Y. led the write-up of the manuscript with inputs from H.T.M., B.L., S.N., J.L., J.H., F.L., R.D., J.W., N.Z., and W.K.

## Competing interests
The authors declare no competing interests.
