## [Transparent Peer Review file · Nature Communications]

Asynchronicity of deglacial permafrost thawing controlled by millennial-scale climate variability

Corresponding Author: Professor Xu Zhang

Version 0:

Reviewer comments:

Reviewer #1

(Remarks to the Author)

Review of the manuscript "Asynchronicity of deglacial permafrost thawing controlled by millennial-scale climate variability" by Yan et al.

This is a manuscript reconstructing permafrost history of Tibetan Plateau (TP) by dating of 600,000 y speleothem record from Tianmen cave system by U-series chronology. The authors find that unlike Circum-Arctic permafrost, TP permafrost initiates thawing early in deglaciations, and they argue that permafrost thaws coincide with sluggish Atlantic Meridional Overturning Circulation (AMOC) during Heinrich stadials. The authors write that weakening of South Asian Summer Monsoon that accompanies the weak AMOC induces anomalous TP warming through local cloud – precipitation - soil moisture feedbacks. This, combined with cooling in Circum-Arctic regions, results in asynchronous permafrost thawing across the Northern Hemisphere. The authors also argue that during the last deglaciation, however, the early recovery of the AMOC during the Bølling-Allerød, coupled with a persistent AMOC during Heinrich Stadial 1, effectively suppresses the local feedbacks, advancing Circum-Arctic while delaying TP permafrost thawing. Authors propose that permafrost carbon release, influenced by millennial-scale AMOC variability, is a non-trivial contributor to deglacial atmospheric CO₂ rise, highlighting the intricate interplay between permafrost dynamics and global climate systems.

I think that this study can be published in Nature Communications if the authors address the following comments:

In cold or dry regions speleothem growth in caves depends on amount of water penetrating into the caves. The latter frequently depends on drainage surface area from which the water seeps into the cave. If the drainage area is relatively small, for example less than one square km, the cave receives its water from precipitation that falls directly on the surface above the cave. However, if there are stream channels above the cave that drain much larger areas, then there is a potential of much larger amounts of water to infiltrate into vadose zone during the summer. This can lead to formation of "taliks" ("holes" or "windows" in the permafrost under the stream channel) through which the water can infiltrate down into the cave. In this case the cave with larger drainage area above it will show more speleothem deposition periods than the cave with small drainage area. In areas of near-zero mean annual temperatures and discontinuous permafrost such local factors can especially important.

Siberian caves of Vaks et al 2013, 2020 have relatively small drainage areas, with no prominent stream channels above the caves.

Therefore, it could be useful if the authors could present a figure showing a map of the research site with the estimation of the approximate area from which the cave receives its water. Is it a hilltop or there is a stream channel found above the cave? How large the drainage area this stream channel, if found?

If the drainage area of Tianmen cave system is particularly large (>1 square km), than the additional thawing periods found in Tianmen cave system compared to the Circum-Arctic caves could be also a result of local factors (amount of water, drainage area), and not only of different climatic behaviour. However, if the drainage area above the cave is small (for example, it drains only the local hilltop), then it is very likely that its permafrost thawing record indeed represents a unique climate behaviour compared to Circum-Arctic region.

(Remarks to the Author)

This is a well-written paper and discussing a very interesting and timely topic. The paper has compiled U/Th ages from speleothems of the Tibetan Plateau published in recent and decade older papers (QSR 2022, 2012) and compared the ages with those of polar permafrost, for which high precision chronology of growth phases in Siberia was published recently (2013, 2020) and lower precision chronology from Canada was also considered. The paper also presents new analysis of climate model simulations to diagnose the effects of AMOC weakenings on temperature on the Tibetan Plateau. The results are well-illustrated in the main and extended figures and the text and methods section are clear. However, from the current presentation, it is not clear if the speleothem age data fully support the interpretations made for some of the interpretations. A more robust statistical treatment of the data and nuanced discussion of the findings is required for the paper to be suitable for publication in Nature Communications.

The robustness of conclusions of timing of growth in Siberia vs Tibet:

Certainly the greater error of the older U-Th dates needs to be taken into account. However, the cutoff of disregarding ages with uncertainty greater than 40 ky (and including those with 28 ky uncertainties) is arbitrary and does affect the conclusions of the study. For example in the Siberian data in TV, discarding two ages with slightly higher uncertainties than the other, results in a significantly younger calculated onset for Siberia than Tibet, but that conclusion would not be born out if all data were considered. A solution which accounts for all the ages and their uncertainty and calculates with robust statistical approach the different onsets of growth phases is required to support the conclusions. The resulting message may be slightly more complex (eg Tibet may not always lead Siberia) but it is important that the paper shows the actual results supported by a transparent and robust synthesis of the data, and undertake more complex interpretations when they are required by the data.

Likewise, in TIV the conclusion of earlier onset of Tibetan permafrost thawing appears to hang on the interpretation of start if “continuous growth” in the Siberia excluding the early dates, but it is meaningful that there was a phase of early Siberian growth. The criteria of “Continuous growth” of polar permafrost speleothems requires greater clarity and perhaps greater nuance for the case of MIS 9e, where a significant Siberian growth phase of >5 ky with many dates is indicated prior to the defined “onset of continuous growth” which may also last only 6 ky (depending if the two significantly younger dates are integrated). Extended Data Figure 6 is really key here and should be cited earlier in the text and include all ages and their uncertainties. There is probably something important to be learned from the early growth phase of Siberia at TIV and its cessation as Tibet growth starts – there is still some antiphasing – can the authors hypothesize the climatic mechanism? The clearest lag of Siberia relative to Tibet growth is seen in TIII, where Siberia misses 7e and begins growth after TIIIa. The lag of Siberia relative to Tibet in TII is clear, but is only 2 ky, which also illustrates how difficult it would be to confirm a similar lag in older terminations TIV and TV where the dating uncertainty is 4 to 6 ky. Overall, further discussion of the differences in relative timing among the terminations would be instructive and provide deeper lessons from the comparison.

The growth rate for the stalagmites derived from the U/Th data is also not presented but where there are multiple dates in a given speleothem (suggested by the sample names in the supplementary table), growth rate might be another important indication of the continuity of permafrost-free conditions on the centennial-scale and the overall intensity of soil respiration which relates to the key carbon cycle feedbacks cited in the introduction.

Can interpretations be made about the conditions at which growth ends in each growth phase and the controlling climatological parameters?

The robustness of comparison with other AMOC records:

Given the uncertainty in the chronology of many of the marine sediment archives of the AMOC over terminations (see next paragraph), I suggest that comparisons with the other absolute dated speleothem record (East Asian Monsoon Composite) need to be highlighted more extensively, especially since recent work clarifies the manifestation of AMOC reductions in the EASM isotope record (He et al., 2021). A more powerful diagnostic of the phasing of Siberian and Tibetan permafrost warming may be their comparison with the absolute dated record of the EASM in which age uncertainties are good and robustly quantified. For TII, a directly dated North Atlantic speleothem record also provides constraints on the timing of AMOC disruption (Stoll et al., 2022) and other Central European records (e.g. Koltai et al., 2017) provide an indication of the last warming step of the termination.

Another concern is the conclusions drawn about the relationship with other events, without discussion of the chronological uncertainties which affect such a comparison. The uncertainty of the new speleothem age models is clearly described and the uncertainty in absolute ages can be readily quantified. However, many of the records employed in the comparison in Figure 2 and the discussion, derive from marine records and/or ice cores which also have significant (and not always well constrained) age uncertainty. To support the interpretations made from comparison with such records, further discussion is required of the marine and ice core age models and their uncertainties from those records – presumably their originally published age models are used – and how the uncertainties in those age models might affect the conclusions made. For example, estimates of the uncertainty for ice core CO₂ records is 6000 years on AICC2012 (Bazin et al 2012). Thus quoting specific ppm CO₂ values for the onset of the Tibetan permafrost thawing over the terminations is not accurate, the uncertainties due to chronology need to be accounted for. A realistic assessment of uncertainties on the marine records used in interpretation is also needed; the benthic δ¹⁸O stack, tuned to insolation, has several ky of uncertainty so the confidence with which one attributes TP melting to “a range of ice volume levels” should also be assessed.

In Extended Data Figure 5 the choice of records for North Atlantic SST is a bit unusual, since for T1 and TII there are Iberian Margin Uk'37 (same proxy as Indian ocean record shown) which more clearly show the cooling during the H1 (and YD) and H11 events. The Iberian margins have been tuned to speleothem chronology in a paper including one of the co-authors of this study (Tzedekis, Drysdale et al Nat. Comm 2018) as well as in Stoll et al., 2022.

The interpretations based on the model

The model interpretation adds significant new perspective of drivers of Tibetan permafrost warming and relationship to the AMOC and Figure 3 is very clear. It would be quite instructive to put the location of the Siberian speleothems on Figure 3, to also highlight that there is no net warming during the hosing at this location. The discussion of mechanisms will likely need to be broadened beyond the AMOC to account for the more diverse phasing between Tibet and Siberia when the U-Th dates are treated in a more transparent and statistically robust way.

Other issues

I am not an expert in the calculations from U-Th system and cannot comment on the appropriateness of the recalculations of published ages with a different ratio for the detrital correction and hope that other reviewers can provide assessment of this.

It is a bit unusual that in Acknowledgments, only the authors of the papers presenting U/Th dates from Tibetan records (Cai, Wang) are thanked for their previous work, but not the author of the Siberian studies (Vaks) which provide a comparable number of records in the compilation. If discussion and feedback about the Tibetan records is what is meant to be acknowledged, then this should be specified. If the authors intend to highlight the significant effort in production of the original dates, then those who produced data on both Tibetan and Siberian regions should be acknowledged.

References cited:

He, Chengfei, Zhenyu Liu, B. L. Otto-Bliesner, E. C. Brady, C. Zhu, R. Tomas, P. U. Clark et al. "Hydroclimate footprint of pan-Asian monsoon water isotope during the last deglaciation." *Science Advances* 7, no. 4 (2021): eabe2611.

Koltai, G., Spötl, C., Shen, C.C., Wu, C.C., Rao, Z., Palcsu, L., Kele, S., Surányi, G. and Bárányi-Kevei, I., 2017. A penultimate glacial climate record from southern Hungary. *Journal of Quaternary Science*, 32(7), pp.946-956.

Stoll, Heather M., Isabel Cacho, Edward Gasson, Jakub Sliwinski, Oliver Kost, Ana Moreno, Miguel Iglesias et al. "Rapid northern hemisphere ice sheet melting during the penultimate deglaciation." *Nature communications* 13, no. 1 (2022): 3819.

Tzedakis, P. C., Russell N. Drysdale, Vasiliki Margari, Luke C. Skinner, Laurie Menviel, Rachael H. Rhodes, Andréa Sardinha Taschetto et al. "Enhanced climate instability in the North Atlantic and southern Europe during the Last Interglacial." *Nature communications* 9, no. 1 (2018): 4235.

Reviewer #3

(Remarks to the Author)

This manuscript compiles previously published speleothem ages from caves in regions with permafrost and compares the growth history of speleothems on the Tibetan Plateau with that of speleothems from circumarctic caves in Siberia and Canada. The authors conclude that speleothem growth in the Tibetan Plateau tends to begin relatively early in deglaciations, during deglacial Heinrich stadials, suggesting early permafrost melting. In contrast, the ages from circumarctic caves (after some filtering) suggest a later onset of growth, suggesting that permafrost melting does not begin in these regions until the glacial termination is nearly complete. The authors also examine climate model output and find that changes in cloud cover and precipitation in response to AMOC weakening produce substantial temperature increases on the Tibetan Plateau, potentially explaining thaw during Heinrich stadials.

This manuscript offers a useful compilation and analysis of existing ages, and it presents an interesting hypothesis. The suggestion that monsoon weakening and surface drying during Heinrich stadials leads to warming of the land surface on the Tibetan Plateau, driving early permafrost melting during deglaciations, seems well supported by model output, though I have some questions below about ways to extend and strengthen this analysis. The data the authors compile is broadly consistent with their hypothesis, but I do have some questions about the robustness of the authors' conclusions about the data and their examination of AMOC changes during terminations.

1. The Tibetan Plateau data show a more mixed relationship with AMOC changes than the authors suggest. The onset of growth in T-V and T-III appears to be within the Weak Monsoon Intervals (WMI) that accompany deglacial AMOC disruptions, but the onset of growth in T-IV is well before that termination's deglacial WMI; the T-II onset of growth is right at the end of the WMI/beginning of the interglacial; and as noted there is no onset of growth during the T-I WMI. The manuscript provides a potential explanation for T-I but does not seem to acknowledge and discuss the weaker association of growth and AMOC disruptions in T-II and T-IV. It also seems worth mentioning that given this variable timing, it seems unlikely that TP permafrost plays a key role in deglacial CO₂ rise.

2. As the Tibetan Plateau dates come from samples with dense dating and age-depth models, why don't the authors use a speleothem age-depth modeling package (e.g., COPRA) rather than just relying on the individual ages? This would be more powerful in demonstrating that the limited ages that provide evidence for their hypothesis are robust and not outliers or out of stratigraphic order, and it would provide better estimates of the onset of deposition during each termination.

3. One key limitation of the study is that the Siberian and Canadian dating was conducted quite differently from the Tibetan Plateau work. The TP dating aimed to develop full age models, and so dated the full thickness of each section of growth. In contrast, only a few dates were collected on most samples from Siberia and Canada, and most dating focused on the outermost deposits to determine the most recent period of growth. As a result, there was far less effort put into determining the onset of growth. This limitation should be acknowledged as a key uncertainty and an area for further work.

4. I'm concerned about the impact of the excluded ages. Many of the older ages from the Siberian and Canadian datasets are on the older end of the age distribution, including many from early MIS11. The authors should explore the impact of a less restrictive filter or no filter on their KDEs and estimates for the onset of growth.

5. The mechanism by which AMOC weakening leads to warming on the plateau is through weakening of the Asian monsoon, as the monsoon changes lead to reduced cloud cover and precipitation. The authors should put more emphasis on comparisons with U/Th dated Asian monsoon reconstructions (e.g., the composite record of Cheng et al., 2016) rather than AMOC and IRD proxies, as these have a less direct relationship to temperature and permafrost changes on the Tibetan Plateau and have much greater age uncertainty. Currently the Cheng et al. reconstruction is shown in Fig 2, but the text focuses on comparisons with records from the North Atlantic. I would argue it is better to focus more attention in the text on comparisons to a record that reflects some part of Asian monsoon variations and is on a U/Th age model that is directly comparable to the speleothem ages presented here.

6. A simple diffusive model for the heat budget of a cave would be useful in assessing the authors' hypothesis. If we take the modeled temperature changes in response to AMOC disruption at face value, would these changes be sufficient to thaw the epikarst from a reasonable glacial temperature, and how long should it take? I'm trying to understand why 2-3 kyr of AMOC weakening in H1 wasn't enough to initiate speleothem growth prior to the Bølling-Allerød. In addition, I'm curious about why rather small temperature changes in the Holocene (e.g. late Holocene cooling, Medieval warmth) were sufficient to start/stop growth, but somehow growth was able to start at the very beginning of T-IV while CO₂ and insolation were still low and prior to the AMOC disruption.

7. I'm still unclear why heat accumulation in the Indian Ocean is important. The authors explain well the direct effects of monsoon weakening through reduced cloud cover and soil moisture. I understand that eastern Indian Ocean warming occurs at the same time, but I don't understand its mechanistic relationship to the local temperature changes on the Tibetan Plateau.

Smaller comments:

83 "the" is unnecessary before "cave carbonate deposits"

182 Change "Less clouds increase" to "Less cloud cover increases"

201-2 Can the authors briefly describe the data and model results cited in this sentence, to help the reader understand what these conclusions are based on?

205 Change to "warming effect of reduced cloud cover"

284 Here and in the data tables and figure captions, I believe "2 " should be changed to "2 ".

634 Why is the focus on boreal summer? Shouldn't the caves be responding to mean annual temperature?

651 This figure should also show the locations of the Canadian and Siberian caves.

658 I'm not sure what the dD record (f) is showing. Also, if North Indian Ocean temperatures are important, why aren't they shown for the most recent deglaciation?

687 "Benthic" is misspelled in the axis labels of multiple figures. Also, again – why isn't the Chinese speleothem d18O record shown here, given that the authors' hypothesis links WMLs to local thaw?

Version 1:

Reviewer comments:

Reviewer #1

(Remarks to the Author)

Review of the manuscript "Asynchronicity of deglacial permafrost thawing controlled by millennial-scale climate variability" by Yan et al.

This is a manuscript reconstructing permafrost history of Tibetan Plateau by dating of 0.5 My speleothem record from Tianmen cave system by U-series chronology. The authors find that unlike Circum-Arctic permafrost, TP permafrost initiates thawing early in deglaciations, coinciding with weak monsoon intervals (WMI) and sluggish Atlantic Meridional Overturning Circulation (AMOC) during Heinrich stadials. The authors write that weakening of South Asian Summer Monsoon induces anomalous TP warming through local cloud – precipitation – soil moisture feedbacks. This, combined with cooling in circum-Arctic regions, results in asynchronous permafrost thawing across the Northern Hemisphere. The authors also argue that during the last deglaciation, however, the early recovery of the AMOC during the Bølling-Allerød, coupled with a persistent AMOC during Heinrich Stadial 1, effectively suppressed the local feedbacks, advancing circum-Arctic permafrost thawing while delaying thawing of permafrost in the TP. Authors argue that basing on their study, permafrost carbon release, influenced by millennial-scale AMOC variability, is a non-trivial contributor to deglacial atmospheric CO₂ rise.

I am generally satisfied with authors reply to my comments in the first review. However, I suggest some corrections to the paragraph (lines 117-133) they modified to address these comments.

I suggest to formulate the first two sentences of the above paragraph as following:

"The cave is located on the plateau few tens of meters above a river. The cave receives its water from direct rain and

snowmelt above it. Absence of stream channels above the cave prevents formation of “Taliks” (“holes” or “windows” in the permafrost under the stream channel) through which the water can infiltrate down into the cave.”

I also don't agree to the following statement:

“Our results show that speleothem growth in Tianmen cave generally commences within the dry and weak monsoon intervals (WMI) associated with a collapsed AMOC state during terminations³¹ (Fig. 2; Extended Data Fig. 2 and 3). This suggests that the growth of speleothem in the study region is not influenced by the availability of meteoric water and runoff, but rather the thawing water from the overlying permafrost caused by the AMOC collapse-induced warming.”

The amount of annual precipitation in the study area is 440 mm, which is much higher than any threshold of speleothem precipitation (200-350 mm, (Vaks et al. 2010)). Therefore it is wrong to argue that the permafrost thaw is the only source of water for the speleothem growth. I say that both atmospheric precipitation and permafrost thaw caused growth of speleothems in the study region.

Reference:

Vaks, A., et al. (2010). "Middle-Late Quaternary paleoclimate of northern margins of the Saharan-Arabian Desert: reconstruction from speleothems of Negev Desert, Israel." *Quaternary Science Reviews* 29(19-20): 2647-2662.

Reviewer #2

(Remarks to the Author)

The authors have made a substantial effort in revisions, and many of the revisions significantly improve the manuscript (while a few others do not)

There were three main issues in the original manuscript which needed to be addressed in the revision

(1) More robust data treatment especially better criteria when some age data are excluded, an aspect significantly affecting the interpretations. This is overall much better resolved in the revision, because now an analysis is provided which includes all of the available dates and Extended Figure 2 shows this clearly. On the other hand, the new definition of “continuous growth phases” which the authors use to neglect the significance of an early Arctic growth phase at MIS 9, still needs better attention. The criteria of continuous being defined by “overlapping error bars on successive dates” is not an ideal criteria because it depends on the error on the dates (depends on age and technique) and depends on the density of sampling for ages (a growth phase may be continuous but sampled at widely spaced intervals so that age uncertainty may not overlap). The current definition appears chosen ad hoc to allow the authors to disregard this early growth phase. It may be better to acknowledge that it is difficult in slow growing stalagmites to verify if growth is continuous but to indicate that for this study a growth phase is defined as an increment of speleothem deposition supported by multiple consecutive ages without evidence for visual discontinuity of growth. Then acknowledge the long Siberian growth phase preceding MIS 9e and address it in Figure 2 and discuss it. Extended data Figure 2 is a very clear and transparent presentation of the data. Figure 2 shows all the dates for Tibet but does not show all the dates for Siberia and Canada, a choice that obscures the complexity of the data. Why not show all the dates for all regions for a transparent presentation of the results in Figure 2 (this would be more important than showing 2 f and 2g which give context but are redundant compared to information in 2e).

2) Better attention to the chronological uncertainties, and the suggestion that the EASM speleothem records be used for comparison since there is strong evidence they record AMOC

Reviewers 2 and 3 both suggested that the authors base their interpretation of timing on comparison with the absolute dated d18O record of east Asian monsoon speleothems, which is documented to record the impact of AMOC on the Asian monsoon system with periods of weak AMOC manifest in millennial positive anomalies traditionally referred to as “weak monsoon intervals”. The authors have incorporated the EASM d18O record as suggested as well as the Tibetan cave d18O with a citation to its link to the south Asian monsoon. It may be worth emphasizing in the text that these d18O records indicate broad changes in the monsoon circulation during AMOC weakenings, but not necessarily changes in the amount of precipitation at the site. (otherwise stating that something is a “summer monsoon proxy” will lead most readers to conclude the d18O is related to the amount of summer monsoon precipitation, yet the speleothem d18O does not have a consistent relationship with precipitation amount across the Asian monsoon domain). Furthermore statements such as “It has been suggested that reductions in Asian Summer Monsoon during Terminations are a consequence of AMOC slow down during terminal stadials” could also be misleading. Rather, than using the word “reduction” in Asian summer monsoon, indicate “Changes in the Asian summer monsoon circulation manifest in positive d18O speleothem anomalies, are a consequence of AMOC slow down during terminal stadials”.

Lines 181-188, added in revision, add some not useful clauses and some misinterpretation. First, D18O speleothem signals record different facets of the climate system in different regions, so d18O carbonate from the Arctic is not necessarily an AMOC indicator (especially in the Canadian Arctic). This first sentence is therefore not coherent with the line of argument and can be deleted. Most importantly the records cited in this paragraph only cover Termination II. This adds some detail to the results already inferred from the east Asian monsoon record d18O chronology but it does not strengthen the interpretation of Tibet with the weak AMOC state in any other terminations and the sentences at the end of the paragraph overstate the importance. The citation of TII records can be added as detail adding a further support in the previous

paragraph, emphasizing that they provide confirmation for TII supporting the timing inferred from East Asian monsoon records. Critically, in this section and the extended figure the authors have misinterpreted the North Atlantic (Iberian) speleothem isotope record. The $\delta^{18}\text{O}$ in NW Iberia is not an AMOC indicator, the $\delta^{13}\text{C}$ from this record is a temperature indicator which is sensitive to the stadial temperature decreases from AMOC weakening, as described in the publication. Therefore the $\delta^{13}\text{C}$, rather than $\delta^{18}\text{O}$, is more relevant for this NW Iberia record in the comparison figure and the text need to be corrected so that it does not allude to the $\delta^{18}\text{O}$ of the North Atlantic cave.

Lines 189-200 do not really address the fundamental concern that the marine records do not provide suitable chronology to add any support to the argument. The first sentence is misleading "In contrast to circumarctic permafrost, the TP permafrost thawing generally coincided with deglacial Asian WMI (i.e. AMOC disruptions) during T-II to T-V. This is further supported by the AMOC proxies derived from marine records"

Comparison with marine records cannot be used to support that TP thawing coincided with AMOC disruptions. Rather, the marine records give a richer context for the phasing of the AMOC disruptions relative to the overall progression of terminations and can be used in this way. It is good that the authors have now described the age models for the marine cores and ice core which is useful. Illustrating the IRD and $\delta^{13}\text{C}$ in Figure 2 f and g is OK to give the context if it does not come at the cost of other more vital information. But nothing new is gained by adding new text about the relationship with the timing of IRD peaks. IRD records come from marine sediment cores which have no absolute chronology. Even when the chronology of the marine record includes a few tuning points to the EASM speleothem $\delta^{18}\text{O}$ record, from the point of view of chronology and timing, nothing more is gained for this study by comparing the phases with an IRD record – it is equivalent to comparing directly with the "weak monsoon events" in the EASM record which is why reviewers 2 and 3 both suggested using the EASM record. This paragraph should be adjusted to emphasize not the chronology and phasing but to enrich the context of the events and highlight that there is a sequence of events and cite the literature which has led to the modern framework for understanding terminations. Something to the effect that:

"Although the chronology for marine records is much less precise than speleothem records, marine records confirm that glacial terminations feature a sequence of changes with increased iceberg rafting and benthic foram $\delta^{13}\text{C}$ evidence for a shallow or weak AMOC at the onset of the termination preceding the main deglaciation as manifest in benthic $\delta^{18}\text{O}$. This has been widely interpreted as early deglacial freshwater addition disrupting AMOC"

3) A key recommendation was more nuance in the interpretation and better attention to the variation in phasing where it is not always possible to document that Tibet leads Arctic thawing given the dating uncertainties.

This needs to be better acknowledged in the paragraph from lines 167 to 180, and lines 210-212 can highlight this better. Cutting unnecessary text in previous sections as suggested in comment 2, will give room to better acknowledge this. The dataset is good and honesty about it will only strengthen the paper and lead to more effective future modeling and data collection. Acknowledge in which terminations it is clear that Tibet leads Arctic responses. (line 170 to 171 say Tibet onset is "generally within deglaciations" with the exception of T1, but this "generalization" should be followed by a statement that clearly indicates in which cases the onset is well constrained to be in the deglaciation, in which cases the onset is likely (within error) of the deglacial stadial/WMI, and in which cases the onset is not (e.g. T1)

The addition of analysis of temperature profiles and permafrost thawing from the model runs was an interesting addition. Reviewer 3 indicated that the case for permafrost CO_2 release was overly substantiated but this is still the first sentence of the abstract, so it may be necessary to revise the abstract.

The revised manuscript contains issues in the English language that should be addressed before the manuscript can be considered for publication. Contractions are not used in formal English, other spelling and grammatical errors should be fixed. Usage such as "We assume this is related to regional change" is not really right for a sentence forwarding a hypothesis tested with a model, rather say "One possible cause is regional change on the Tibetan plateau..." or "our favored explanation is ". Also usage such as "We want to highlight that..." is not stylistically used. Better, "It is noteworthy that..." "The author list includes coauthors whose first language is English should be able to address these and other issues if they take time to read the manuscript carefully.

Reviewer #3

(Remarks to the Author)

I appreciate the revisions the authors have made, and in particular the addition of the TM $\delta^{18}\text{O}$ records in Supp. Fig 3 and the heat diffusion modeling. I have three further requests for revisions.

First, the last sentence of the abstract is too strong. This sentence states that the authors show that permafrost carbon release "is a non-trivial contributor to deglacial atmospheric CO_2 rise." The authors show that permafrost carbon may have contributed to deglacial atmospheric CO_2 rise, but they do not show that it is "non-trivial", as they do not provide constraints on how much carbon was released from permafrost. I suggest changing to "may have been a non-trivial contributor to deglacial atmospheric CO_2 rise."

Second, the new text added to the manuscript has multiple small grammatical errors that should be fixed prior to publication. Review of this new text by all co-authors should be sufficient to address these errors.

Third, the first section of the methods needs more information about the cave. How deep is the cave? Is it relatively horizontal, or does it slant downwards/upwards? Approximately how thick is the epikarst above the cave?

Version 2:

Reviewer comments:

Reviewer #1

(Remarks to the Author)

I see that the last sections that needed correction were modified. Therefore, I think this manuscript is ready for publication.

Reviewer #2

(Remarks to the Author)

Reviewer #3

(Remarks to the Author)

I am satisfied by the authors' revisions, with the exception that line 30 should be changed to, "Permafrost is a potentially important source of deglacial carbon release alongside..."

REVIEWER COMMENTS

Reviewer #1 (Remarks to the Author):

Review of the manuscript “Asynchronicity of deglacial permafrost thawing controlled by millennial-scale climate variability” by Yan et al.

1. This is a manuscript reconstructing permafrost history of Tibetan Plateau (TP) by dating of 600,000 y speleothem record from Tianmen cave system by U-series chronology. The authors find that unlike Circum-Arctic permafrost, TP permafrost initiates thawing early in deglaciations, and they argue that permafrost thaws coincide with sluggish Atlantic Meridional Overturning Circulation (AMOC) during Heinrich stadials. The authors write that weakening of South Asian Summer Monsoon that accompanies the weak AMOC induces anomalous TP warming through local cloud – precipitation - soil moisture feedbacks. This, combined with cooling in Circum-Arctic regions, results in asynchronous permafrost thawing across the Northern Hemisphere. The authors also argue that during the last deglaciation, however, the early recovery of the AMOC during the Bølling-Allerød, coupled with a persistent AMOC during Heinrich Stadial 1, effectively suppresses the local feedbacks, advancing Circum-Arctic while delaying TP permafrost thawing. Authors propose that permafrost carbon release, influenced by millennial-scale AMOC variability, is a non-trivial contributor to deglacial atmospheric CO₂ rise, highlighting the intricate interplay between permafrost dynamics and global climate systems. I think that this study can be published in Nature Communications if the authors address the following comments:

R1: We appreciate referee #1’s positive appraisal on our manuscript as well as the concerns on potential factors for speleothem growth in cold regions.

2. In cold or dry regions speleothem growth in caves depends on amount of water penetrating into the caves. The latter frequently depends on drainage surface area from which the water seeps into the cave. If the drainage area is relatively small, for example less than one square km, the cave receives its water from precipitation that falls directly on the surface above the cave. However, if there are stream channels above the cave that drain much larger areas, then there is a potential of much larger amounts of water to infiltrate into vadose zone during the summer. This can lead to formation of “taliks” (“holes” or “windows” in the permafrost under the stream channel) through which the water can infiltrate down into the cave. In this case the cave with larger drainage area above it will show more speleothem deposition periods than the cave with small drainage area. In areas of near-zero mean annual temperatures and discontinuous permafrost such local factors can especially important. Siberian caves of Vaks et al 2013, 2020 have relatively small drainage areas, with no prominent stream channels above the caves. Therefore, it could be useful if the authors could present a figure showing a map of the research site with the estimation of the approximate area from which the cave receives its water. Is it a hilltop or there is a stream channel found above the cave? How large the drainage area this stream channel, if found? If the drainage area of Tianmen cave system is particularly large (>1 square km), than the additional thawing periods found in Tianmen cave system compared to the Circum-Arctic caves could be also a

result of local factors (amount of water, drainage area), and not only of different climatic behaviour. However, if the drainage area above the cave is small (for example, it drains only the local hilltop), then it is very likely that its permafrost thawing record indeed represents a unique climate behaviour compared to Circum-Arctic region.

R2: We agree with the reviewer that larger drainage areas could provide more substantial water infiltration into the vadose zone, potentially leading to the formation of "taliks" and enhanced speleothem deposition. According to geologic survey, the entrance of Tianmen Cave is on the hilltop while a stream channel originated from lake Nam Co flows beside the hill and likely below the cave system (Fig. R1). Thus, the primary source of water for Tianmen Cave is meteoric precipitation, rather than the stream infiltration. Nonetheless, no matter the water is supplied by precipitation or stream, the main water source of Tianmen Cave is held back in the permafrost system, and its amount reaching the cave is modulated by temperature regimes controlling the thawing process of the permafrost above the cave.

Figure R1. The stream channel above the Tianmen Cave.

Additionally, if the water availability was the key to the growth of Tibetan speleothem, it would be expected that the initial speleothem growth occurred during the strong monsoon periods with abundant water supply from rainfall. However, the available $\delta^{18}\text{O}_{\text{sp}}$ records from Tianmen cave indicate that speleothem growth generally begins within the Weak Monsoon Interval (WMI) associated with a collapsed AMOC during terminations (Fig. R2) (Cheng et al., 2016). In fact, the growth could occur during both weak (e.g., MIS 5e, 7e, 9e) and strong monsoon periods (e.g., MIS 5a, 5c, 9a). This is different from the condition in Kesang Cave in arid central Asia (northwestern China), where the speleothem growth is mainly controlled by moisture changes, because the growth generally begin with strong monsoon period. Consequently, we believe the climatic warming in the study region is the first-order control on the speleothem growth by thawing the overlying permafrost and warming cave.

We have revised the discussion, please see **lines 117-133**.

“The cave is located on the plateau besides a small creek, which prevent formation of “Taliks” (“holes” or “windows” in the permafrost under the stream channel) through which the water can infiltrate down into the cave. The growth of speleothems in permafrost regions first requires the thawing of the permafrost which is equivalent to deepening the active layer. Rain- or meltwater from ground ice can then subsequently enter the cave through soil percolation. Here, we use uranium-thorium dating of speleothem calcite during active growth periods, and pair it with $\delta^{18}\text{O}$ on the same samples to identify whether TP speleothems grew during rain-heavy (i.e., strong monsoon), or dry (i.e., weak monsoon) periods across 5 terminations during the last 500 ka. We then used kernel density estimation (KDE)^{26–28} to generate probability distributions of our compiled ages for TP^{25,29,30} and published circumarctic speleothems from Siberia and Canada^{21–24} (Fig. 1g, h, i) to reconstruct active speleothem growth periods over the past ~500 ka. Our results show that speleothem growth in Tianmen cave generally commences within the dry and weak monsoon intervals (WMI) associated with a collapsed AMOC state during terminations³¹ (Fig. 2; Extended Data Fig. 2 and 3). This suggests that the growth of speleothem in the study region is not influenced by the availability of meteoric water and runoff, but rather the thawing water from the overlying permafrost caused by the AMOC collapse-induced warming.”

Figure R2. From top to down, summer insolation precession; Asian Monsoon composite record; Tianmen cave $\delta^{18}\text{O}$ record, composed by six stalagmites in different colors; Kesang cave $\delta^{18}\text{O}$ record; Sea level reconstruction.

Replotted from Wang et al. (2022).

Ref.

Wang, H., Wang, X., Pérez-Mejías, C., Li, Y., Li, H., Cai, Y., Zhang, H., Han, J., Duan, P., Lu, J., Ning, Y., Edwards, R.L., Cheng, H., 2022. Orbital-scale hydroclimate variations in the southern Tibetan Plateau over the past 414,000 years. *Quaternary Science Reviews* 291, 107658.

Cheng, H., Edwards, R.L., Sinha, A., Spötl, C., Yi, L., Chen, S., Kelly, M., Kathayat, G., Wang, X., Li, X., Kong, X., Wang, Y., Ning, Y., Zhang, H., 2016. The Asian monsoon over the past 640,000 years and ice age terminations. *Nature* 534, 640–646. <https://doi.org/10.1038/nature18591>

Reviewer #2 (Remarks to the Author):

1. This is a well-written paper and discussing a very interesting and timely topic. The paper has compiled U/Th ages from speleothems of the Tibetan Plateau published in recent and decade older papers (QSR 2022, 2012) and compared the ages with those of polar permafrost, for which high precision chronology of growth phases in Siberia was published recently (2013, 2020) and lower precision chronology from Canada was also considered. The paper also presents new analysis of climate model simulations to diagnose the effects of AMOC weakenings on temperature on the Tibetan Plateau. The results are well-illustrated in the main and extended figures and the text and methods section are clear. However, from the current presentation, it is not clear if the speleothem age data fully support the interpretations made for some of the interpretations. A more robust statistical treatment of the data and nuanced discussion of the findings is required for the paper to be suitable for publication in Nature Communications.

R1: We thank referee #2 for the supportive and critical comments on our work. In the following we provide the point-to-point responses to the comments.

2. The robustness of conclusions of timing of growth in Siberia vs Tibet:

Certainly the greater error of the older U-Th dates needs to be taken into account. However, the cutoff of disregarding ages with uncertainty greater than 40 ky (and including those with 28 ky uncertainties) is arbitrary and does affect the conclusions of the study. For example in the Siberian data in TV, discarding two ages with slightly higher uncertainties than the other, results in a significantly younger calculated onset for Siberia than Tibet, but that conclusion would not be born out if all data were considered. A solution which accounts for all the ages and their uncertainty and calculates with robust statistical approach the different onsets of growth phases is required to support the conclusions. The resulting message may be slightly more complex (eg Tibet may not always lead Siberia) but it is important that the paper shows the actual results supported by a transparent and robust synthesis of the data, and undertake more complex interpretations when they are required by the data.

R2: We appreciate your concerns on the means we used which might undermine the robustness of our conclusions, especially for Termination V when the U-Th method approaches its limits. In the new analysis, we included all the dates which are shown in Extended Data Fig. 1, 2. The new results are in line with our original conclusions, as elaborated below.

Previously, we filtered out records with age uncertainties greater than 40 kyr when comparing the onsets of continuous thawing periods between Tibetan Plateau (TP) and circumarctic

permafrost region. The U-Th dates with age certainty greater than 40 kyr accounting for 13% of the dataset (Figure R3), most of which ages are older than 400 ka. This filtering approach thus mainly influences the results of Termination V when U-Th methods reaching the dating limits. Part of Canadian alpha spectrometry dates were also disregarded as there exists large uncertainties in their earlier dates, which even indicate speleothem growth during glacial periods when temperature is cold enough to avoid substantial permafrost thawing. A possible explanation is that these samples were collected from cave entrances, which are subject to seasonal snowmelt. Nevertheless, including these alpha spectrometry dates does not alter our conclusions.

Figure R3. The distribution of all age uncertainty (2σ) of Tianmen Cave samples.

In the new analysis, we include all the dates (including alpha spectrometry dates) without any data filtration (Figure R4). We acknowledged that the increased uncertainty associated with the older U-Th dates complicate the comparison among different records during T-V. For instance, the uncertainty range can span the whole interglacial or glacial periods. This to some extent downgrades the robustness of other valuable and reliable finely-dated samples, reducing their potential to investigate the initial timing of the speleothem growth. Taking together we consider that the dates with higher precision can provide more reliable information about timing. Therefore, the dates with relative lower dating uncertainties were given more weight and considered as the most possible timing for the onset of persistent permafrost thawing (Methods). Based on this criterion, the onset of Tibetan Plateau permafrost thawing remains earlier than (> 8 ky) circumarctic permafrost during Termination V. Overall, the results based on the synthesis of all the dates (incl. earlier alpha spectrometry dates) from past four Terminations remain supporting the original conclusions.

Figure R4. Selection of the dates representing onset of continuous thawing during interglacial MIS 1 to MIS 11c. The panels show the distribution of Siberian (blue), Canadian (purple) and TP (red) speleothem ages ($\pm 2\sigma$) in time. The relatively large bold circle with error bars indicates the onset of continuous permafrost thawing in circumarctic (i.e. Canada and Siberia) and TP (Methods). The upper panel background is the East Asian Summer Monsoon composite records² (green). The grey error bars of T-VI represent a temporary growth event. Vertical bars indicate the deglacial Heinrich stadials.

3. Likewise, in TIV the conclusion of earlier onset of Tibetan permafrost thawing appears to hang on the interpretation of start if “continuous growth” in the Siberia excluding the early dates, but it is meaningful that there was a phase of early Siberian growth. The criteria of “Continuous growth” of polar permafrost speleothems requires greater clarity and perhaps greater nuance for the case of MIS 9e, where a significant Siberian growth phase of >5 ky with many dates is indicated prior to the defined “onset of continuous growth” which may also last only 6 ky (depending if the two significantly younger dates are integrated). Extended Data Figure 6 is really key here and should be cited earlier in the text and include all ages and their uncertainties. There is probably something important to be learned from the early growth phase of Siberia at TIV and its cessation as Tibet growth starts – there is still some antiphasing – can the authors hypothesize the climatic mechanism?

R3: Thanks for pointing this out. “Continuous growth” here only refers to a continuous speleothem growth period without prominent interruptions whilst with the primary growth period occurring during warm interglacial periods. Only such “continuous growth” can represent the thawing of permafrost caused by warming. The growth interruptions mean no overlap between the two adjacent dates even if the age uncertainties are taken into account. The primary growth should occur mainly during warm interglacial periods as suggested by the established causality between speleothem growth and overlying ground thawing. Please see **Methods** for details.

“Criteria for continuous growth and selection of the dates representing onset of continuous thawing

“Continuous growth” here only refers to a continuous speleothem growth period without prominent interruptions, with the primary growth period occurring during warm interglacial periods. Only such “continuous growth” can represent the thawing of permafrost caused by warming. The growth interruptions mean no overlap between the two adjacent dates even if the age uncertainties are taken into account. The primary growth should occur during warm interglacial periods as suggested by the established causality between speleothem growth and overlying ground thawing (Extended Data Fig. 2).

The dates with too large uncertainties (e.g. spanning the whole interglacial or glacial periods) would undermine other valuable and reliably finely dated samples, making them worthless for providing useful information when investigating the initial timing of growth. It is reasonable that the dates with lower uncertainty can provide more valuable information. Therefore, the dates with the earliest timing and relative low dating uncertainties were assumed as the most possible timing for the onset of persistent permafrost thawing.

The Siberian speleothem “growth phase” during MIS 10 was separated from later interglacial “continuous growth” phase during MIS 9e. The cause for this glacial growth event before the termination remains unclear to us. It may be not driven by climatic warming as demonstrated in low lake Baikal biogenic silica contents during MIS 10 (Extended Data Fig. 12). This early growth phase is identified by 4 dates from 4 individual speleothems. In addition to their age uncertainty, we argue that these dates are more likely to indicate a brief growth event rather than a growth stage. For example, a brief warm event may cause a significant deepening of the active layer of permafrost during the summer, leading to the transient growth of speleothem. Therefore, this early growth was not considered as a “continuous growth” in our study. Due to the complexity of the permafrost thawing mechanisms, we acknowledge additional work is needed to further clarify the driving mechanisms of this early growth event, if existing.”

During T-IV, it is noteworthy that there is a ~5 ky Siberian speleothem growth phase (343.2-347.9 ka BP), which was identified by 4 dates. In contrast, the “continuous growth” of Siberian speleothem during MIS 9e lasts for ~13 ky (313.5-326.7 ka BP), as identified by 10 dates. This is much longer than the preceding temporary “growth phase” of 5 kyr during Termination IV. In general, the interglacial “continuous growth” of Siberian speleothem is shorter than that of the TP speleothem. For instance, the Siberian permafrost thawing during MIS 5, MIS 7 and MIS 9 is limited to parts of the warm Marine Isotope sub-stages, while TP permafrost thawing is nearly persistent through the entire interglacial periods. This might be

associated with warmer climate background over TP than in the circum-Arctic region, indicating that TP permafrost is more vulnerable than circumpolar permafrost under warm climate. This is probably because low latitudes TP permafrost region has thicker active layer compared with circumpolar permafrost due to the higher mean annual air temperature (Wang et al., 2022), which increase the sensitivity of TP permafrost to the thawing, i.e, TP permafrost region require shorter warming duration to deepening the active layer and enable water penetrate vadose zone to the cave system.

The glacial “growth phase” during MIS 10 was separated from later interglacial “continuous growth” phase during MIS 9e. The cause for this glacial growth event before the termination remains unclear to us. It may be not driven by climatic warming as demonstrated in low lake Baikal biogenic silica contents during MIS 10 (Fig. R5). This early growth phase is identified by 4 dates from 4 individual speleothems (SOP-1-D (II); SB-p7497-2-Obot; SOP-18-G; SOP-16-F). In addition to their age uncertainty, we argue that these dates are more likely to indicate a brief growth event rather than a growth stage. For example, a brief warm event may cause a significant deepening of the active layer of permafrost during the summer, leading to the transient growth of speleothem. Additionally, the available pictures of these speleothems reveal that most of these dates (SB-p7497-2-Obot; SOP-18-G; SOP-16-F) were sampled closed to the hiatus, which indicates the samples bear risks by incorporating a small amount of ‘contaminating’ material from the other side of the hiatus, affecting the reliability of the dating results. Therefore, this early growth was not considered as a “continuous growth” in our study. Due to the complexity of the permafrost thawing mechanisms, we acknowledge additional work is needed to further clarify the driving mechanisms of this early growth event, if existing.

Extended Data Fig. 6 has been revised and all ages are shown, we cited this key figure earlier now (Extended Data Fig. 2).

Figure R5. Comparison of speleothems' growth periods in Siberia with benthic stack $\delta^{18}\text{O}$ record and Lake Baikal biogenic silica contents (Prokopenko et al., 2001).

Ref.

Galaasen, E.V., Ninnemann, U.S., Kessler, A., Irvah, N., Rosenthal, Y., Tjiputra, J., Bouttes, N., Roche, D.M., Kleiven, H. (Kikki) F., Hodell, D.A., 2020. Interglacial instability of North Atlantic Deep Water ventilation. *Science* 367, 1485–1489.

Prokopenko, A. A. et al. Biogenic Silica Record of the Lake Baikal Response to Climatic Forcing during the Brunhes. *Quaternary Research* 55, 123–132 (2001).

Wang, X. et al. Contrasting characteristics, changes, and linkages of permafrost between the Arctic and the Third Pole. *Earth-Sci. Rev.* 230, 104042 (2022).

4. The clearest lag of Siberia relative to Tibet growth is seen in TIII, where Siberia misses 7e and begins growth after TIIIa. The lag of Siberia relative to Tibet in TII is clear, but is only 2 ky, which also illustrates how difficult it would be to confirm a similar lag in older terminations TIV and TV where the dating uncertainty is 4 to 6 ky. Overall, further discussion

of the differences in relative timing among the terminations would be instructive and provide deeper lessons from the comparison.

R4: Given the climatic background (global ice volume, greenhouse gases level, insolation) are different during each terminations and interglacials, it is reasonable to have different lead-lag feature among each Terminations. During the Termination III, the onset of continuous Siberian permafrost thawing (~213 ka) lags that of TP (~249 ka) by 36 kyr. This large time lag is a consequence of that the circumarctic speleothem growth skipped MIS 7e and began growing until MIS 7a-c. The average atmospheric CO₂ level during MIS 7e (although there was a high CO₂ spike right after TIII) is lower than MIS 5e, MIS 9e and MIS 11c, suggesting a persistent cooler climatology during this sub-stage, this is also supported by lower biogenic silica contents in Lake Baikal during MIS 7e (Fig. R5). Therefore, permafrost thawing is limited (active layer of permafrost is generally thin), leaving low chances for persistent speleothem growth in MIS7e. The persistent and longer warm periods indicated by CO₂ level of MIS 7a-c compared with MIS 7e finally cause the widespread permafrost thawing in circumarctic region. In contrast, the general warm background in low latitudes results in a thicker active layer in TP compared with circumarctic permafrost (Wang et al., 2022), increase the sensitivity of TP permafrost to the warming, i.e, TP permafrost region require shorter warming duration to thaw the layer and enable water penetrate vadose zone to the cave system. This accounts for its persistent thawing and hence speleothem growth throughout MIS 7a-e. Moreover, during Termination IV, the lag of Siberian (~327 ka) relative to TP permafrost thawing (~347 ka) is about 20 ka after ruling out the earlier transient growth event in Siberia (**R3**). The lead-lag relationship in Termination V is also evident when applied the criterion mentioned in **R2** – the lag of circumarctic permafrost thawing to that in TP is larger than ~8 kyr (Fig. R3). We have revised this, please see **lines 201-212.** and **Methods lines 414-441.**

“Given varying characteristics of the past 5 Terminations, we acknowledge the complex relationship of the TP permafrost thawing with changes in summer insolation, ice volume, and Greenhouse gases (GHGs). For instance, the onset of continues thawing does not align with changes in local summer insolation during deglaciations – TP permafrost thawing occurs during both high (T-II and T-III) or low (T-IV and T-V) rates and magnitudes of summer insolation increase (Fig. 2b). Additionally, there is no obvious relationship between permafrost thawing and GHGs levels, since during T-II and T-V speleothem growth commences during final deglacial CO₂ rising, while during T-III and T-IV speleothem growth onset occurs during the initial stages (Fig. 2c). This is in line with TP permafrost thawing dynamics, which can also occur within a wide range of ice volumes and CO₂ levels (Fig. 2).

Accordingly, it is reasonable to have different lead-lag feature between TP and circumarctic permafrost thawing among each Terminations (Methods).”

“Given the climatic background (global ice volume, greenhouse gases level, insolation) are different during each termination and interglacial, it is reasonable to have different lead-lag feature between TP and circumarctic permafrost thawing among each Terminations. For instance, during T-II, the lag of Siberian permafrost thawing (~128 ka) relative to TP (~130 ka) during T-II is at least ~2 ka, and the lag of Canadian permafrost thawing is even larger. During T-III, the onset of Siberian thawing (~213 ka) lags the TP case (~249 ka) by about 36 kyr. This large time lag is a consequence of that the circumarctic speleothem growth occurred at MIS7a-c by skipping MIS 7e. The average atmospheric CO₂ level during MIS 7e is lower than MIS 5e, MIS 9e and MIS 11c, suggesting a lower climatology air temperature during this interglacial sub-stage, corroborated by lower biogenic silica concentration in Lake Baikal (Extended Data Fig. 12). Therefore, discontinuous/sporadic permafrost and shallower active layer may dominate circumarctic permafrost cave systems, and the chances for growing speleothem under permafrost is low during MIS 7e. During MIS 7a-c, the relatively warmer climate as indicated by ice core CO₂ records, might give rise to the persistent permafrost thawing in circumarctic region. In contrast, the general warm background in low latitudes results in deeper active layer in TP, which increases the sensitivity of TP permafrost to the warming, i.e, TP permafrost region require shorter warming duration to deepen the active layer, which effectively enables thawing water to penetrate vadose zones to the cave system. This accounts for the persistent speleothem growth throughout MIS 7a-e. During T-IV, the lag of Siberian (~327 ka) relative to TP permafrost thawing (~347 ka) is about 20 ka after ruling out a preceded temporary growth event in Siberia. (Methods; Extended Data Fig. 2). The cause for this growth event before the termination is unclear, although it might be not driven by climatic warming as suggested by the low lake Baikal biogenic silica contents during MIS 10 (Extended Data Fig. 12). Therefore, we do not consider this growth event as a part of the continuous permafrost thawing during T-IV. The greater error of the older U-Th dates making it difficult to do the comparison during T-V. Nonetheless, it is reasonable that the dates with lower uncertainty can provide more valuable information. Therefore, the dates with the earliest timing and relative low dating uncertainties were assumed as the most possible timing for the onset of continuous permafrost thawing (Methods). Based on this criterion, we suggest that the lag of circumarctic permafrost thawing to that in TP is larger than ~8 kyr, quantitatively in consistency with T-II and T-IV (Extended Data Fig. 2).”

Ref.

Wang, X. et al. Contrasting characteristics, changes, and linkages of permafrost between the Arctic and the Third Pole. *Earth-Sci. Rev.* 230, 104042 (2022).

5. The growth rate for the stalagmites derived from the U/Th data is also not presented but where there are multiple dates in a given speleothem (suggested by the sample names in the supplementary table), growth rate might be another important indication of the continuity of permafrost-free conditions on the centennial-scale and the overall intensity of soil respiration which relates to the key carbon cycle feedbacks cited in the introduction.

R5: Thanks for this valuable suggestion. Here we show the growth rate for the well laminated and densely dated stalagmites 19TM-2, 19TM-3 and 19TM-5 (Tianmen Cave), which was originally published in Wang et al. (2020); other samples are not suitable for calculating growth rate due to small number of dating points and many growth hiatuses. It is noteworthy that the growth rate was relatively high at the onset of growth during Terminations (Fig. R6), which cannot be explained by the increased monsoon rainfall, as the growth begins during Weak Monsoon Interval (WMI). This high growth rate was probably caused by the abrupt thawing that effectively increases the water amount into the dry cave. Moreover, the permafrost thawing will enhance respiration of previously ice-locked soil carbon, as demonstrated by the decreased stalagmites $\delta^{13}\text{C}$ value during terminations (Fig. R6). This may increase soil $p\text{CO}_2$ level, leading to increased Ca^{2+} concentration in the drip. This will give rise to a higher growth rate (Lechleitner et al., 2021), and indicates an enhanced permafrost carbon release after initial thawing. We have revised the discussion, please see **lines 213-221**.

“In addition, TP speleothem growth rate was generally higher at the onset of growth during deglaciations compared with the ensuing interglacial periods (Extended Data Fig. 5), this cannot be explained by the lower precipitation during Terminations. Instead, this high growth rate again suggests abrupt thawing of ground ice, rather than precipitation, as the primary water source for speleothem growth. Moreover, the permafrost thawing will enhance respiration of previously frozen soil carbon, as demonstrated by the decreased stalagmites $\delta^{13}\text{C}$ value at the onset of thawing³⁸ (Extended Data Fig. 5). The release of such old carbon may enhance $p\text{CO}_2$ levels in the soil leading to increased Ca^{2+} concentration in the drip for higher speleothem growth rate³⁸.”

Figure R6. Plots of age versus depth for stalagmites 19TM-2 (black), 19TM-3 (blue) and 19TM-5 (pink) from Tianmen cave and grow rate time series. The Tianmen $\delta^{13}\text{C}$ record was shown in the bottom panel. Replotted from Wang et al. (2020).

Ref.

Lechleitner, F.A., Day, C.C., Kost, O., Wilhelm, M., Haghypour, N., Henderson, G.M., Stoll, H.M., 2021.

Stalagmite carbon isotopes suggest deglacial increase in soil respiration in western Europe driven by temperature change. *Climate of the Past* 17, 1903–1918. <https://doi.org/10.5194/cp-17-1903-2021>

6. Can interpretations be made about the conditions at which growth ends in each growth phase and the controlling climatological parameters?

R6: The speleothem growth ends as it transitions from interglacial to glacial periods (Fig. 2), in addition to temporary growth stagnations in between the sub-stages (e.g. between MIS 5c and MIS 5e). They are a result of local cooling associated with decreases in the greenhouse gases and local summer insolation and expansion of continental ice sheets, which effectively

reduces summer thawing and ultimately promote permafrost to grow (e.g. shoaling permafrost active layer). Moreover, the growth generally ends during interglacial strong monsoon periods, minimizing influences of rainfall-induced drainage water on speleothem growth.

7. The robustness of comparison with other AMOC records:

Given the uncertainty in the chronology of many of the marine sediment archives of the AMOC over terminations (see next paragraph), I suggest that comparisons with the other absolute dated speleothem record (East Asian Monsoon Composite) need to be highlighted more extensively, especially since recent work clarifies the manifestation of AMOC reductions in the EASM isotope record (He et al., 2021). A more powerful diagnostic of the phasing of Siberian and Tibetan permafrost warming may be their comparison with the absolute dated record of the EASM in which age uncertainties are good and robustly quantified.

R7: Thanks for these very useful suggestions. We have extensively highlighted the comparisons of growth phases with the absolute dated speleothem $\delta^{18}\text{O}$ record (East Asian Monsoon Composite) in the discussion as well as in the figures, which enhance the robustness of conclusions. It has been deemed that a collapsed Atlantic Meridional Overturning Circulation (AMOC) leads to a cold anomaly over the North Atlantic, weakening Asian Summer Monsoon (ASM) by associated atmospheric teleconnections (He et al., 2021), which is supported by the enriched $\delta^{18}\text{O}$ values in the composited Chinese speleothem composite records (Cheng et al., 2016). This physically-based causality provided a robust diagnostic of the timing and phasing of Siberian and Tibetan permafrost thawing. In total 169 sub-samples from 30 individual speleothems from Tianmen Cave were dated, in which 6 carefully selected speleothems were analyzed for their oxygen isotopic ratios (Wang et al., 2022) (Fig. R7). These available $\delta^{18}\text{O}$ records from Tianmen Cave speleothem again indicates the growth starts within Weak Monsoon Interval (in indicated by enriched $\delta^{18}\text{O}$ values during T-II to T-VI) when AMOC collapse. Importantly, this line of evidence is not subject to date uncertainty. In contrast, the growth of circumarctic speleothems is corresponding to the strong monsoon interval during interglacial period (Fig. 2; Figure R8), lagging behind the TP speleothem growth.

We have revised the discussion, please see **lines 167-200**.

“We found that, in contrast to the continuous thawing of circumarctic permafrost, which starts only during warm interglacial periods, the onset of the TP permafrost thawing is generally within deglaciations (with an exception of the last deglaciation, i.e. T-I) (Fig. 2). We assume that this is related to the regional climatic change during deglaciations on the Tibetan Plateau. In general, a common feature is that the commencement of TP permafrost thawing broadly began within the Weak Monsoon Interval (WMI), when AMOC collapsed^{32,34}, as demonstrated in higher $^{18}\text{O}_{\text{carbonate}}$ value of EASM composite (Fig. 2) as well as Tianmen cave $\delta^{18}\text{O}_{\text{carbonate}}$ records (Southern Asian Summer Monsoon proxy)²⁶ (Extended Data Fig. 3). It has been suggested that reductions in Asian Summer Monsoon during Terminations are a consequence of AMOC slow down during terminal stadials^{32,34}. This further indicates that, independent of absolute dates, Tibetan permafrost thawing and deglacial AMOC reductions are likely associated.

Unfortunately, there are currently no available $\delta^{18}\text{O}_{\text{carbonate}}$ records for circumarctic caves. Nonetheless, the $\delta^{18}\text{O}_{\text{carbonate}}$ record of North Atlantic Cave³⁵ and Central European Cave³⁶ provide good constraints on the timing of AMOC disruption and the last warming stage of the T-II (Extended Data Fig. 4). By comparing our onset dates of continuous growth with the absolute dated $\delta^{18}\text{O}_{\text{carbonate}}$ record from the high-latitude regions, we found that circumarctic permafrost thawing does not occur during deglacial AMOC collapse, but rather during the following warm interglacial period. This also suggests that Tibetan permafrost thawing likely occurred earlier than permafrost thawing in the circumarctic regions.

In contrast to circumarctic permafrost, the TP permafrost thawing generally coincided with deglacial Asian WMI (i.e. AMOC disruptions) during T-II to T-V. This is further supported by the AMOC proxies derived from marine records. As such, the onset of TP thawing is accompanied by ice-rafted debris (IRD) events and occurred in tandem with depleted bottom water $\delta^{13}\text{C}$ values from benthic foraminifera in North Atlantic. IRD events occur during the onset/early stage of deglaciations and are a result of freshwater input to the North Atlantic from collapsing northern hemisphere ice sheets³⁷. These “Heinrich Events”³⁷ (Fig. 2g) have been shown to cause the collapse of AMOC by affecting the density of deep waters at North Atlantic deep water convection sites²¹. The reduced production of North Atlantic Deep Water (NADW) and the AMOC disruption can be seen in the records of benthic foraminifera $\delta^{13}\text{C}$ values (Fig. 2f). Our data highlight that the onset of TP speleothem continuous growth occurred during Heinrich events and are thus likely associated with a collapsed AMOC.”

Figure R7. Comparison of Tianmen cave records with East Asian Summer Monsoon composite records. From top to down, insolation precession; East Asian Summer Monsoon composite records; Tianmen cave $\delta^{18}\text{O}$ record, composed by six stalagmites in different colors; ages with errors of Tianmen cave stalagmites; LR04 benthic $\delta^{18}\text{O}$ stack.

Ref.

- He, C., Liu, Z., Otto-Bliesner, B.L., Brady, E.C., Zhu, C., Tomas, R., Clark, P.U., Zhu, J., Jahn, A., Gu, S., Zhang, J., Nusbaumer, J., Noone, D., Cheng, H., Wang, Y., Yan, M., Bao, Y., 2021. Hydroclimate footprint of pan-Asian monsoon water isotope during the last deglaciation. *Science Advances* 7, eabe2611.
- Cheng, H., Edwards, R.L., Sinha, A., Spötl, C., Yi, L., Chen, S., Kelly, M., Kathayat, G., Wang, X., Li, X., Kong, X., Wang, Y., Ning, Y., Zhang, H., 2016. The Asian monsoon over the past 640,000 years and ice age terminations. *Nature* 534, 640–646.
- Wang, H., Wang, X., Pérez-Mejías, C., Li, Y., Li, H., Cai, Y., Zhang, H., Han, J., Duan, P., Lu, J., Ning, Y., Edwards, R.L., Cheng, H., 2022. Orbital-scale hydroclimate variations in the southern Tibetan Plateau over the past 414,000 years. *Quaternary Science Reviews* 291, 107658.

8. For TII, a directly dated North Atlantic speleothem record also provides constraints on the timing of AMOC disruption (Stoll et al., 2022) and other Central European records (e.g. Koltai et al., 2017) provide an indication of the last warming step of the termination.

R8: Thanks for the suggestion. The directly dated North Atlantic speleothem record and Central European speleothem records during Termination II, corroborate our conclusion that Siberian permafrost did not thaw until the start of warm interglacial period, while TP permafrost started thawing in the deglaciation as AMOC slowed down (Fig. R8). We added this to the discussion, see **lines 181-188**.

“Unfortunately, there are currently no available $\delta^{18}O_{\text{carbonate}}$ records for circumarctic caves. Nonetheless, the $\delta^{18}O_{\text{carbonate}}$ record of North Atlantic Cave³⁴ and Central European Cave³⁵ provide good constraints on the timing of AMOC disruption and the last warming stage of the T-II (Extended Data Fig. 4). By comparing our onset dates of continuous growth with the absolute dated $\delta^{18}O_{\text{carbonate}}$ record from the high-latitude regions, we found that circumarctic permafrost thawing does not occur during deglacial AMOC collapse, but rather during the following warm interglacial period. This also suggests that Tibetan permafrost thawing likely occurred earlier than permafrost thawing in the circumarctic regions.”

Figure R8. Comparison of speleothem growth phase North Atlantic (Stoll et al., 2022), Central European (Koltai et al., 2017), and Chinese speleothem composite records (Cheng et al., 2016), also shown is the global climatic background. From top to bottom: Global benthic $\delta^{18}\text{O}$ stack (Lisiecki and Raymo, 2005); Atmospheric CO_2 concentration (Lüthi et al., 2008); Synthetic Greenland $\delta^{18}\text{O}$ record (Barker et al., 2011); Eirik Drift IRD records (Galaasen et

al., 2020) [percent of >150 μm entities]. Red vertical bars indicate the deglacial AMOC disruptions.

Ref:

Barker, S., Knorr, G., Edwards, R.L., Parrenin, F., Putnam, A.E., Skinner, L.C., Wolff, E., Ziegler, M., 2011. 800,000 Years of Abrupt Climate Variability. *Science* 334, 347–351.

Cheng, H., Edwards, R.L., Sinha, A., Spötl, C., Yi, L., Chen, S., Kelly, M., Kathayat, G., Wang, X., Li, X., Kong, X., Wang, Y., Ning, Y., Zhang, H., 2016. The Asian monsoon over the past 640,000 years and ice age terminations. *Nature* 534, 640–646.

Galaasen, E.V., Ninnemann, U.S., Kessler, A., Irvani, N., Rosenthal, Y., Tjiputra, J., Bouttes, N., Roche, D.M., Kleiven, H. (Kikki) F., Hodell, D.A., 2020. Interglacial instability of North Atlantic Deep Water ventilation. *Science* 367, 1485–1489.

Koltai, G., Spötl, C., Shen, C. -C., Wu, C. -C., Rao, Z., Palcsu, L., Kele, S., Surányi, G., Bárány-Kevei, I., 2017. A penultimate glacial climate record from southern Hungary. *J Quaternary Science* 32, 946–956. <https://doi.org/10.1002/jqs.2968>

Lisiecki, L.E., Raymo, M.E., 2005. A Pliocene-Pleistocene stack of 57 globally distributed benthic $\delta^{18}\text{O}$ records. *Paleoceanography* 20.

Lüthi, D., Le Floch, M., Bereiter, B., Blunier, T., Barnola, J.-M., Siegenthaler, U., Raynaud, D., Jouzel, J., Fischer, H., Kawamura, K., Stocker, T.F., 2008. High-resolution carbon dioxide concentration record 650,000–800,000 years before present. *Nature* 453, 379–382.

Stoll, H.M., Cacho, I., Gasson, E., Sliwinski, J., Kost, O., Moreno, A., Iglesias, M., Torner, J., Perez-Mejias, C., Haghypour, N., Cheng, H., Edwards, R.L., 2022. Rapid northern hemisphere ice sheet melting during the penultimate deglaciation. *Nat Commun* 13, 3819.

9. Another concern is the conclusions drawn about the relationship with other events, without discussion of the chronological uncertainties which affect such a comparison. The uncertainty of the new speleothem age models is clearly described and the uncertainty in absolute ages can be readily quantified. However, many of the records employed in the comparison in Figure 2 and the discussion, derive from marine records and/or ice cores which also have significant (and not always well constrained) age uncertainty. To support the interpretations made from comparison with such records, further discussion is required of the marine and ice

core age models and their uncertainties from those records – presumably their originally published age models are used – and how the uncertainties in those age models might affect the conclusions made. For example, estimates of the uncertainty for ice core CO₂ records is 6000 years on AICC2012 (Bazin et al 2012). Thus quoting specific ppm CO₂ values for the onset of the Tibetan permafrost thawing over the terminations is not accurate, the uncertainties due to chronology need to be accounted for. A realistic assessment of uncertainties on the marine records used in interpretation is also needed; the benthic δ¹⁸O stack, tuned to insolation, has several ky of uncertainty so the confidence with which one attributes TP melting to “a range of ice volume levels” should also be assessed.

R9: Thanks for pointing this out and we acknowledge that the previous narrative of our results did not clearly state how we interpret age models from different records. In the revision, we focus on the comparison of our growth phases with ASM records, especially the Weak Monsoon Intervals during deglaciations, and do not quote specific CO₂ concentrations for the onset of permafrost thawing. We believe that this is an improved approach, because the age model of the ASM record can directly be compared to our age model through the δ¹⁸O of speleothem calcite (Extended Data Fig. 3). Despite other climate records may have relative large age uncertainties, it can serve as an important climatic background. Here we focus on the processes during Terminations, where the rapid transitions in climate phases improve age accuracy of marine and ice core records (i.e. AMOC, ice-volume proxies and atmospheric CO₂ level) compared to interglacial or glacial periods, this enhances the reliability of comparisons with absolute dated (i.e., permafrost thawing and Asian monsoon proxies) records (Cheng et al., 2009, 2016). Moreover, in order to better compare the absolute dated record (growth phase and ASM) with marine, ice core records and stacked benthic δ¹⁸O, we used the ASM-tuned chronologies of EDC CO₂ (i.e., by synchronizing the abrupt ASM change with the CH₄ jump) and stacked benthic δ¹⁸O (i.e., chronology tuned during TV), details can be found in Cheng et al. (2009, 2016). We use the original chronologies of benthic δ¹³C and IRD records published in Galaasen et al. (2020), which are well-constrained during Terminations (Fig. R10). As for the AMOC proxies, the age model of *C. wuellerstorfi* δ¹³C records is constrained by tuning to benthic δ¹⁸O stack (Fig. R10), which clearly indicates a deglacial AMOC collapse. Importantly, the onset of TP speleothem growth was within the range of Weak Monsoon Interval (WMI), which is an unambiguously reflection for AMOC collapse and unaffected by age uncertainty.

We have clarified this in the discussion, please see section of “Onset of TP permafrost thawing during deglacial Heinrich stadials” (lines 166-212) and **R6**.

Figure R9. Comparison of Tibetan speleothem growth phase with insolation changes.

Figure R10. Age model correlation for *C. wuellerstorfi* $\delta^{13}\text{C}$ (‰) records.

Ref.

Cheng, H., Edwards, R.L., Broecker, W.S., Denton, G.H., Kong, X., Wang, Y., Zhang, R., Wang, X., 2009. Ice Age Terminations. *Science* 326, 248–252.

Cheng, H., Edwards, R.L., Sinha, A., Spötl, C., Yi, L., Chen, S., Kelly, M., Kathayat, G., Wang, X., Li, X., Kong, X., Wang, Y., Ning, Y., Zhang, H., 2016. The Asian monsoon over the past 640,000 years and ice age terminations. *Nature* 534, 640–646.

Galaasen, E.V., Ninnemann, U.S., Kessler, A., Irvál, N., Rosenthal, Y., Tjiputra, J., Bouttes, N., Roche, D.M., Kleiven, H. (Kikki) F., Hodell, D.A., 2020. Interglacial instability of North Atlantic Deep Water ventilation. *Science* 367, 1485–1489.

Parrenin, F., Barnola, J.-M., Beer, J., Blunier, T., Castellano, E., Chappellaz, J., Dreyfus, G., Fischer, H., Fujita, S., Jouzel, J., Kawamura, K., Lemieux-Dudon, B., Loulergue, L., Masson-Delmotte, V., Narcisi, B., Petit, J.-R., Raisbeck, G., Raynaud, D., Ruth, U., Schwander, J., Severi, M., Spahni, R., Steffensen, J.P., Svensson, A., Udisti, R., Waelbroeck, C., Wolff, E., 2007. The EDC3 chronology for the EPICA Dome C ice core. *Climate of the Past* 3, 485–497.

Lisiecki, L.E., Raymo, M.E., 2005. A Pliocene-Pleistocene stack of 57 globally distributed benthic $\delta^{18}\text{O}$ records. *Paleoceanography* 20.

10. In Extended Data Figure 5 the choice of records for North Atlantic SST is a bit unusual, since for T1 and TII there are Iberian Margin Uk'37 (same proxy as Indian ocean record shown) which more clearly show the cooling during the H1 (and YD) and H11 events. The Iberian margins have been tuned to speleothem chronology in a paper including one of the co-authors of this study (Tzedekis, Drysdale et al Nat. Comm 2018) as well as in Stoll et al., 2022.

R10: Revised, please see Extended Data Fig. 10.

11. The interpretations based on the model:

The model interpretation adds significant new perspective of drivers of Tibetan permafrost warming and relationship to the AMOC and Figure 3 is very clear. It would be quite instructive to put the location of the Siberian speleothems on Figure 3, to also highlight that there is no net warming during the hosing at this location. The discussion of mechanisms will likely need to be broadened beyond the AMOC to account for the more diverse phasing between Tibet and Siberia when the U-Th dates are treated in a more transparent and statistically robust way.

R11: Revised. The location of the Siberian speleothems was denoted on Figure 3. All the U-Th dates was considered to investigate the phasing between Tibetan and Siberian permafrost thawing. Please see **R2** and **Methods**.

12. Other issues

I am not an expert in the calculations from U-Th system and cannot comment on the appropriateness of the recalculations of published ages with a different ratio for the detrital correction and hope that other reviewers can provide assessment of this. It is a bit unusual that in Acknowledgments, only the authors of the papers presenting U/Th dates from Tibetan records (Cai, Wang) are thanked for their previous work, but not the author of the Siberian studies (Vaks) which provide a comparable number of records in the compilation. If discussion and feedback about the Tibetan records is what is meant to be acknowledged, then this should be specified. If the authors intend to highlight the significant effort in production

of the original dates, then those who produced data on both Tibetan and Siberian regions should be acknowledged.

R12: Thanks for these comments and we revised the acknowledgement accordingly. All the authors presenting valuable speleothems dates in permafrost regions are highly appreciated. Please see **lines 702-706**.

“Acknowledgements

We thank Anton Vaks, Haibo Wang, Yanjun Cai, Nicole Biller-Celander, Russell Harmon and their co-authors for the exhaustive studies of Cave speleothems in permafrost region.”

References cited:

- He, C., Liu, Z., Otto-Bliesner, B.L., Brady, E.C., Zhu, C., Tomas, R., Clark, P.U., Zhu, J., Jahn, A., Gu, S., Zhang, J., Nusbaumer, J., Noone, D., Cheng, H., Wang, Y., Yan, M., Bao, Y., 2021. Hydroclimate footprint of pan-Asian monsoon water isotope during the last deglaciation. *Science Advances* 7, eabe2611.
- Koltai, G., Spötl, C., Shen, C. -C., Wu, C. -C., Rao, Z., Palcsu, L., Kele, S., Surányi, G., Bárányi-Kevei, I., 2017. A penultimate glacial climate record from southern Hungary. *J Quaternary Science* 32, 946–956.
- Stoll, H.M., Cacho, I., Gasson, E., Sliwinski, J., Kost, O., Moreno, A., Iglesias, M., Torner, J., Perez-Mejias, C., Haghypour, N., Cheng, H., Edwards, R.L., 2022. Rapid northern hemisphere ice sheet melting during the penultimate deglaciation. *Nat Commun* 13, 3819.
- Tzedakis, P.C., Drysdale, R.N., Margari, V., Skinner, L.C., Menviel, L., Rhodes, R.H., Taschetto, A.S., Hodell, D.A., Crowhurst, S.J., Hellstrom, J.C., Fallick, A.E., Grimalt, J.O., McManus, J.F., Martrat, B., Mokeddem, Z., Parrenin, F., Regattieri, E., Roe, K., Zanchetta, G., 2018. Enhanced climate instability in the North Atlantic and southern Europe during the Last Interglacial. *Nat Commun* 9, 4235.

Reviewer #3 (Remarks to the Author):

This manuscript compiles previously published speleothem ages from caves in regions with permafrost and compares the growth history of speleothems on the Tibetan Plateau with that of speleothems from circumarctic caves in Siberia and Canada. The authors conclude that speleothem growth in the Tibetan Plateau tends to begin relatively early in deglaciations, during deglacial Heinrich stadials, suggesting early permafrost melting. In contrast, the ages from circumarctic caves (after some filtering) suggest a later onset of growth, suggesting that permafrost melting does not begin in these regions until the glacial termination is nearly complete. The authors also examine climate model output and find that changes in cloud cover and precipitation in response to AMOC weakening produce substantial temperature increases on the Tibetan Plateau, potentially explaining thaw during Heinrich stadials.

This manuscript offers a useful compilation and analysis of existing ages, and it presents an interesting hypothesis. The suggestion that monsoon weakening and surface drying during Heinrich stadials leads to warming of the land surface on the Tibetan Plateau, driving early permafrost melting during deglaciations, seems well supported by model output, though I have some questions below about ways to extend and strengthen this analysis. The data the authors compile is broadly consistent with their hypothesis, but I do have some questions about the robustness of the authors' conclusions about the data and their examination of AMOC changes during terminations.

Re: We thank referee #3 for the recognition of the quality and importance of our work.

1. The Tibetan Plateau data show a more mixed relationship with AMOC changes than the authors suggest. The onset of growth in T-V and T-III appears to be within the Weak Monsoon Intervals (WMI) that accompany deglacial AMOC disruptions, but the onset of growth in T-IV is well before that termination's deglacial WMI; the T-II onset of growth is right at the end of the WMI/beginning of the interglacial; and as noted there is no onset of growth during the T-I WMI. The manuscript provides a potential explanation for T-I but does not seem to acknowledge and discuss the weaker association of growth and AMOC disruptions in T-II and T-IV. It also seems worth mentioning that given this variable timing, it seems unlikely that TP permafrost plays a key role in deglacial CO₂ rise.

R1: Based on the available $\delta^{18}\text{O}$ records from Tianmen Cave, the onsets of speleothem growth in T-II, T-III and T-IV were well constrained within the Weak Monsoon Intervals (WMI) that accompany with deglacial AMOC disruptions (Figure R11).

Figure R11. Comparison of Tianmen cave records with East Asian Summer Monsoon composite records. From top to down, insolation precession; East Asian Summer Monsoon composite records; Tianmen cave $\delta^{18}\text{O}$ record, composed by six stalagmites in different colors; ages with errors of Tianmen cave stalagmites; LR04 benthic $\delta^{18}\text{O}$ stack.

By comparing with the absolute dated Chinese composed speleothem records, the T-II onset of TP speleothem growth is at the end of the WMI and hence in the middle of the termination, while the Siberian speleothem growth occurred until the beginning of the interglacial (indicated by high EASM composite $\delta^{18}\text{O}$ records). In addition, the onset of T-IV growth was also within the WMI when taking into account the relatively large dating uncertainty (Fig. R7) and the growth onset in T-V is also alongside with WMI (Fig. R12). Collectively, growth onsets in T-II to T-IV are within the WMIs (after AMOC collapse) and earlier than onset of interglacial periods. We have added a section discuss on the asynchronicity of Siberian and TP permafrost thawing during each deglaciation. Four out of five terminations in the last 450kyr, the TP permafrost continuous thawing occurred within the WMI during early

deglaciation, earlier than circumarctic permafrost. In the manuscript, we do not claim that TP permafrost plays a key role in deglacial CO₂ rise. We suggest that the Tibetan permafrost thawing is a potential carbon source for deglacial CO₂ rise, especially within deglacial stadials (i.e. WMI). Nonetheless, we have revised the expression of the significance of TP permafrost carbon thawing. We have revised the discussion, for the condition in T-II and T-IV please see **lines 173-188** and **Methods lines 431-441**.

“In general, a common feature is that the commencement of TP permafrost thawing broadly began within the Weak Monsoon Interval (WMI), when AMOC collapsed^{32,34}, as demonstrated in higher $\delta^{18}O_{\text{carbonate}}$ value of EASM composite (Fig. 2) as well as Tianmen cave $\delta^{18}O_{\text{carbonate}}$ records (Southern Asian Summer Monsoon proxy)²⁶ (Extended Data Fig. 3). It has been suggested that reductions in Asian Summer Monsoon during Terminations are a consequence of AMOC slow down during terminal stadials^{32,34}. This further indicates that, independent of absolute dates, Tibetan permafrost thawing and deglacial AMOC reductions are likely associated.

Unfortunately, there are currently no available $\delta^{18}O_{\text{carbonate}}$ records for circumarctic caves. Nonetheless, the $\delta^{18}O_{\text{carbonate}}$ record of North Atlantic Cave³⁵ and Central European Cave³⁶ provide good constraints on the timing of AMOC disruption and the last warming stage of the T-II (Extended Data Fig. 4). By comparing our onset dates of continuous growth with the absolute dated $\delta^{18}O_{\text{carbonate}}$ record from the high-latitude regions, we found that circumarctic permafrost thawing does not occur during deglacial AMOC collapse, but rather during the following warm interglacial period. This also suggests that Tibetan permafrost thawing likely occurred earlier than permafrost thawing in the circumarctic regions.”

“During T-IV, the lag of Siberian (~327 ka) relative to TP permafrost thawing (~347 ka) is about 20 ka after ruling out a preceded temporary growth event in Siberia. (Methods; Extended Data Fig. 2). The cause for this growth event before the termination is unclear, although it might be not driven by climatic warming as suggested by the low lake Baikal biogenic silica contents during MIS 10 (Extended Data Fig. 12). Therefore, we do not consider this growth event as a part of the continuous permafrost thawing during T-IV. The greater error of the older U-Th dates making it difficult to do the comparison during T-V. Nonetheless, it is reasonable that the dates with lower uncertainty can provide more valuable information. Therefore, the dates with the earliest timing and relative low dating uncertainties were assumed as the most possible timing for the onset of continuous permafrost thawing (Methods). Based on this criterion, we suggest that the lag of circumarctic permafrost thawing to that in TP is larger than ~8 kyr, quantitatively in consistency with T-II and T-IV (Extended Data Fig. 2).”

Figure R12. Comparison of the onset of Tibetan speleothems growth with East Asian Summer Monsoon composite records.

2. As the Tibetan Plateau dates come from samples with dense dating and age-depth models, why don't the authors use a speleothem age-depth modeling package (e.g., COPRA) rather than just relying on the individual ages? This would be more powerful in demonstrating that the limited ages that provide evidence for their hypothesis are robust and not outliers or out of stratigraphic order, and it would provide better estimates of the onset of deposition during each termination.

R2: The dense dating results ($n = 169$) from Tianmen Cave system are derived from 30 individual speleothem, most of the stalagmites have only a few dating points because of growth hiatuses. We fully agree with the reviewer that age-depth profiles would be beneficial. Regrettably, the few data points per stalagmite are not suitable for establishing age-depth model.

3. One key limitation of the study is that the Siberian and Canadian dating was conducted quite differently from the Tibetan Plateau work. The TP dating aimed to develop full age models, and so dated the full thickness of each section of growth. In contrast, only a few dates were collected on most samples from Siberia and Canada, and most dating focused on the outermost deposits to determine the most recent period of growth. As a result, there was

far less effort put into determining the onset of growth. This limitation should be acknowledged as a key uncertainty and an area for further work.

R3: Thanks for pointing out this difference. We note that the samples for U-Th analyses in Siberian stalagmites were taken as close as possible to growth hiatuses, and were mainly used to assess the timing of onset and cessation of speleothem growth (Vaks et al., 2013) (Fig. R13). Similarly, the dating for Canadian sample was also focus on the top layer and basal layer of the stalagmites (Harmon et al., 1977).

Figure R13. Major speleothems from Siberian Caves, with their ages in ka (Vaks et al., 2013; 2020).

While stalagmites from the Tibetan Plateau were dated without deliberate selection of the boundary samples, but very close to it. This will lead to even slightly earlier thawing of TP permafrost than the dates presented, which further supports the lag of circumarctic permafrost thawing relative to TP. We have acknowledged this differences and limitations. See **lines 373-381**.

“It should be point out that the sub-samples for U-Th analyses in Tibetan speleothems were taken without deliberate selection of the boundary samples, but very close to it. While the most dating in circumarctic samples were taken as close as possible to growth hiatuses (sampling at a hiatus risks incorporating a small amount of ‘contaminating’ material from the other side of the hiatus, therefore samples were taken a small distance from hiatuses). This will lead to even slightly earlier thawing of TP permafrost than the dates presented. This limitation need more further work to inprove, while it does not change our conclusions. Moreover, the earlier deglacial growth of TP speleothems is an unambiguous phenomenon that occurred during all investigated terminations except T-I (i.e. T-II to T-V), indicating that this early growth is nonrandom and unaffected by artificial sampling biases.”

Ref.

Vaks, A., Gutareva, O.S., Breitenbach, S.F.M., Avirmed, E., Mason, A.J., Thomas, A.L., Osinzev, A.V., Kononov, A.M., Henderson, G.M., 2013. Speleothems Reveal 500,000-Year History of Siberian Permafrost. *Science* 340, 183–186.

Vaks, A., Mason, A.J., Breitenbach, S.F.M., Kononov, A.M., Osinzev, A.V., Rosensaft, M., Borshevsky, A., Gutareva, O.S., Henderson, G.M., 2020. Palaeoclimate evidence of vulnerable permafrost during times of low sea ice. *Nature* 577, 221–225.

Harmon, R.S., Ford, D.C., Schwarcz, H.P., 1977. Interglacial chronology of the Rocky and Mackenzie Mountains based upon ^{230}Th – ^{234}U dating of calcite speleothems. *Can. J. Earth Sci.* 14, 2543–2552.

4. I'm concerned about the impact of the excluded ages. Many of the older ages from the Siberian and Canadian datasets are on the older end of the age distribution, including many from early MIS11. The authors should explore the impact of a less restrictive filter or no filter on their KDEs and estimates for the onset of growth.

R4: In the initial version, we did not apply any filter on KDE. In the revised manuscript, we did not perform any date filtration, all the dates (including alpha spectrometry dates) were taken into account (Extended Data Fig.1, 2) and the conclusion remains the same. For details, please refer to reply to **referee #2-R2** and **R7**.

5. The mechanism by which AMOC weakening leads to warming on the plateau is through weakening of the Asian monsoon, as the monsoon changes lead to reduced cloud cover and precipitation. The authors should put more emphasis on comparisons with U/Th dated Asian monsoon reconstructions (e.g., the composite record of Cheng et al., 2016) rather than AMOC and IRD proxies, as these have a less direct relationship to temperature and permafrost changes on the Tibetan Plateau and have much greater age uncertainty. Currently the Cheng et al. reconstruction is shown in Fig 2, but the text focuses on comparisons with records from the North Atlantic. I would argue it is better to focus more attention in the text on comparisons to a record that reflects some part of Asian monsoon variations and is on a U/Th age model that is directly comparable to the speleothem ages presented here.

R5: Thanks for the important suggestion. We have now focus on the comparison with the absolute dated record of the EASM composite in which age uncertainties are good and robustly quantified, and the Figure 2 and Extended Fig. 2 was revised to strengthen such direct comparison. We further compared the TP dates with $\delta^{18}\text{O}$ records of Tianmen Cave itself and revised the discussion. See **referee #1 R7** and **Section 'Onset of TP permafrost thawing during deglacial Heinrich stadials' (lines 166-229)**

6a. A simple diffusive model for the heat budget of a cave would be useful in assessing the authors' hypothesis. If we take the modeled temperature changes in response to AMOC disruption at face value, would these changes be sufficient to thaw the epikarst from a reasonable glacial temperature, and how long should it take?

R6a: We thank the reviewer for this suggestion which helped to improve our findings. We have included a plot (Figure R14) and a brief description on methods about the expected temperature diffusion patterns into the permafrost in the supplement.

We used the widely published and applied permafrost heat diffusion model as defined in Ballantyne (2018) and in the equation below:

$$T_z = T_a + A_s * e^{-z \sqrt{\frac{\pi}{K_a * p}}}$$

where T_z is the resulting temperature at depth z , T_a is the annual mean surface temperature, A_s is the annual mean surface temperature amplitude defined as annual max temperature + annual mean temperature, K_a is the diffusivity of the ground above the cave, and p is the period of time reflecting the temperature parameters (here annual, 365 days). The resulting plots (Figure R14) show the depth of the permafrost active layer, where $T_z > 0^\circ\text{C}$ (Figure R14). The active layer defines the depth to which the ground thaws and re-freezes on an annual basis, while the underlying ground stays frozen (permafrost). As such, the underlying permafrost can only thaw due to significant and prolonged climate warming.

Figure R14. Permafrost temperature plots for three global climate scenarios. (a) the Holocene, (b) the Last Glacial Maximum (LGM) with a glacial AMOC, and (c) during a Heinrich Event in the glacial setting where AMOC is collapsed.

As evident from the equation above, the permafrost diffusion curves are highly dependent on the temperature information that feeds into the equation. We therefore collected annual temperature data from our model runs (Extended Data Table 1; Figure R15), and compared the temperature output from the Holocene “piCTRL” to published temperature information from the Tibetan Plateau.

Fig R15. Simulated temperatures on Tianmen Cave in Pre-industrial (piCTRL), the Last Glacial Maximum (CTRL) and the LGM hosing (Hosing) experiments. Summer warming anomaly in Hosing is evident in comparison to CTRL.

Extended Data Table 1: Ground surface temperature information for the Tibetan Plateau from model runs presented in this study

Ground Surface Temperature	Annual [°C]	Min [°C]	Max [°C]
piCTRL	-0.69	-13.03	7.90
CTRL	-5.63	-18.08	4.80
Hosing	-4.85	-17.24	6.31

Our modelled Holocene annual temperature amplitude of $-13.03\text{ °C} - 7.90\text{ °C}$, and annual average of 0.69 °C , for the Tibetan Plateau are in good agreement with the annual amplitude of -12 °C to 8 °C , and annual averages across sites of -1 °C to -4 °C published by Wu & Zhang (2008). After choosing a representative thermal diffusivity (K_a) of $0.02\text{ mm}^2/\text{s}$, we

receive a comparable trumpet temperature curve to the measured curve by Wu & Zhang (2008), with a similar Holocene active layer depth of 5 m.

Next, we identified the average annual surface temperature and average annual surface temperature amplitude from the model runs for “LGM + AMOC off (hosing)”, and “LGM + AMOC on (CTRL)”, and used them to calculate the diffusion curves for those climates (Figure R14 b, c). As expected, our Figure R14 shows that the permafrost was in general colder and more stable during the glacial maximum compared to the Holocene, and that the active layer was shallower. Comparing the “hosing” experiment with the “CTRL” experiment highlights that the different AMOC conditions cause a small shift in the temperature regimes leading to a slightly higher mean annual surface temperature, and a greater mean annual surface temperature amplitude in the “hosing” compared to the “CTRL” experiment. The equation above, as well as the trumpet curves, show that such a shift in the temperature regime will inevitably lead to a deepening of the active layer and a thawing of the adjacent permafrost. Based on field observation, water dripping is very active in the cave. This suggests that the cave ceiling is currently located within the active layer (i.e. shallower than 5m), rather than permafrost. Our ages from the last glacial maximum suggest no growth prior to changes in the AMOC. We take this as evidence that the cave ceiling must have been located deeper than 2m which is our modelled active layer depth for the LGM. The regional temperature shift after AMOC change (i.e. hosing vs CTRL experiments) would be sufficient to deepen the active layer by 30cm on the Tibetan Plateau which is substantial in terms of thawed ground volume, and which must have impacted the cave system. In combination with our age data, we suggest that this has a significant impact on the amount of water entering the cave and commencing speleothem growth.

REDACTED

We have added above new analysis to the Methods, and in discussion.

“Using a permafrost heat diffusion model (Methods), we confirm that the AMOC-induced TP warming during terminal stadials can effectively deepen the permafrost active layer, promoting its thawing (Extended Data Fig. 8). As a result, associated meltwater would penetrate into the cave for speleothem growth. ”

Ref.

Ballantyne, C.K., 2018. Periglacial geomorphology. John Wiley & Sons, pp.41-46

Wu, Q. and Zhang, T., 2008. Recent permafrost warming on the Qinghai-Tibetan Plateau. Journal of Geophysical Research: Atmospheres, 113(D13).

R6b. I’m trying to understand why 2-3 kyr of AMOC weakening in H1 wasn’t enough to initiate speleothem growth prior to the Bølling-Allerød.

R6b: There are growing lines of evidence that AMOC was weak but still active during Heinrich Stadial 1 (Oppo et al., 2015; Pöppelmeier et al., 2023; Repschläger et al., 2021). This active AMOC state cannot give rise to robust warming to cause the TP permafrost thawing during the early stage of T-I. Although there is a lack of comparative studies of AMOC strength across past five deglaciations, the carbon isotopes of benthic foraminifera in the North Atlantic (ODP Site 983) do suggest a more active AMOC strength during T-I comparing with previous four Terminations (Extended Data Fig. 11). Moreover, it has been found that AMOC slowdown will lead to Indian Ocean SST warming, and the Indian Ocean SST warming is more attenuated in T-I than that in T-II suggests a more active AMOC during Heinrich stadial 1 (Extended Data Fig. 10), again suggests a more active AMOC during T-I.

Further analysis is needed on how long the shutdown of AMOC would take to cause TP permafrost thawing. According to our modeling in **R6a**, the underlying permafrost can only thaw due to significant and prolonged climate warming. During H1, the active AMOC and B/A interruption event failed to cause the significant warming and would not be sufficient to deepen the active layer and cause the growth of speleothem.

We have revised the discussion, please see **lines 265-292**.

“Of particular interest is that TP permafrost thawing did not occur until the Holocene, while circumarctic permafrost experienced an earlier thawing during the last deglaciation (Fig. 2). This is consistent with previous circumarctic permafrost studies of marine sediments^{16,17} and model simulations^{5,6}, suggesting a remobilization and release of permafrost carbon during last deglaciation. Compared to previous Terminations, the Bølling-Allerød event⁴⁷, a unique feature of T-I, effectively dampened the warming effect of reduced cloud cover due to the rapid recovery of AMOC during the middle stage of the deglaciation. In contrast, this abrupt recovery of AMOC led to a deglacial thawing of circumarctic permafrost.

The onset of TP permafrost thawing generally occurred at the middle or end of weak monsoon intervals (WMI) during T-II to T-V (Fig. 2), suggesting that the early AMOC disruption in Heinrich Stadial 1 was unable to deepen the active layer and cause the growth of speleothem due to insufficient heat accumulation. Moreover, there are growing lines of evidence that AMOC was relative active during Heinrich Stadial 1 (ref. 48–50) and the Younger Dryas⁵¹, although weaker than its state during the Last Glacial Maximum^{49,52}, and the mean ITCZ shift is less than 1° latitude during Heinrich Stadial 1 (ref. 53). We therefore suggest that the weaker but still active AMOC during Heinrich Stadial 1 cannot give rise to robust warming and hence delayed the TP permafrost thawing during T-I. Despite the lack of comparative studies on the AMOC activity during different Terminations, it has been demonstrated that a weakened AMOC can produce anomalous warming in the Indo-Pacific ocean^{54,55} (Extended Data Fig. 9). The records show that Indian Ocean SST warming relative to the North Atlantic is more attenuated in T-I than T-II, it may suggest that the AMOC is more active during T-I than T-II (Extended Data Fig. 10). Also, carbon isotope ratios of benthic foraminifera in North Atlantic (ODP Site 983) suggest a more active AMOC strength during T-I compared with the previous four terminations (Extended Data Fig. 11). These observations strengthen the crucial role of AMOC change in determining the onset of permafrost thawing during deglaciation. Therefore, the evident phase difference in onset timing of permafrost thawing between circumarctic and TP (Methods) can be well explained by changes in the strength of the AMOC that exerts opposing thermal effects in the two regions (Fig. 4). ”

Ref.

Oppo, D.W., Curry, W.B., McManus, J.F., 2015. What do benthic $\delta^{13}\text{C}$ and $\delta^{18}\text{O}$ data tell us about Atlantic circulation during Heinrich Stadial 1? *Paleoceanography* 30, 353–368.

Pöppelmeier, F., Jeltsch-Thömmes, A., Lippold, J., Joos, F., Stocker, T.F., 2023. Multi-proxy constraints on Atlantic circulation dynamics since the last ice age. *Nat. Geosci.*

Repschläger, J., Zhao, N., Rand, D., Lisiecki, L., Muglia, J., Mulitza, S., Schmittner, A., Cartapanis, O., Bauch, H.A., Schiebel, R., Haug, G.H., 2021. Active North Atlantic deepwater formation during Heinrich Stadial 1. *Quaternary Science Reviews* 270, 107145.

R6c. In addition, I'm curious about why rather small temperature changes in the Holocene (e.g. late Holocene cooling, Medieval warmth) were sufficient to start/stop growth, but somehow growth was able to start at the very beginning of T-IV while CO₂ and insolation were still low and prior to the AMOC disruption.

R6c: The Holocene temperature change event may cause a change in the continuity of the overlying permafrost, resulting in stalagmites to grow/stop in localized areas in the cave. For example, the TP permafrost continuously thawed during warm MIS 5e, but individual stalagmites (shown by different color) only growth part of MIS 5e (Fig. R16), which may be a response to interglacial temperature fluctuations that altered the permafrost condition (i.e. the thickness of the active layer) overlaying the specific stalagmites. Given the limited stalagmite samples (2 samples) in MIS 1, it is also possible that speleothem elsewhere growing throughout the Holocene are not collected. Nonetheless, the existing 2 speleothem samples support that the TP stalagmite didn't begin to grow until the onset of the Holocene.

As for T-IV, when taken into account the age uncertainty, the growth was start within the WMI with AMOC disruption (Extended Data Fig. 2), which is also support by the $\delta^{18}\text{O}$ records of Tianmen Cave itself (Extended Data Fig. 3).

We have revised the discussion, please see **lines 155-165**.

“The discontinuities in Holocene (MIS 1) speleothem growth of TP may be a response to interglacial temperature fluctuations (growth ceased due to the cooling after Holocene Climatic Optimum and regrowth during Medieval Warming period; Extended Data Fig. 2) that altered the permafrost condition, and the thickness of active layer, overlaying the specific stalagmites. This results in speleothems to grow/stop growing in localized areas in the cave. For example, the TP permafrost continuously thawed during warm MIS 5e, but individual speleothems only grew during parts of MIS 5e (ref. ²⁶). Given the limited stalagmite samples

(2 samples) collected in MIS 1, it is also possible that speleothem elsewhere growing throughout the Holocene are not collected. Nonetheless, the existing 2 speleothem samples support that the TP stalagmite didn't begin to grow until the onset of the Holocene.”

Figure R16. Tianmen cave $\delta^{18}\text{O}$ record, composed by six stalagmites in different colors.

7. I'm still unclear why heat accumulation in the Indian Ocean is important. The authors explain well the direct effects of monsoon weakening through reduced cloud cover and soil moisture. I understand that eastern Indian Ocean warming occurs at the same time, but I don't understand its mechanistic relationship to the local temperature changes on the Tibetan Plateau.

R7: Sorry for the lack of clarity. We compared the SST changes in North Atlantic and Indian Ocean during T-I and T-II, the records show that Indian Ocean SST warming relative to the North Atlantic is more attenuated in T-I than T-II, suggesting that the AMOC is more active during T-I than T-II (Extended Data Fig. 11)

We have revised the discussion about this. Please see **lines 281-292**.

“Despite the lack of comparative studies on the AMOC activity during different Terminations, it has been demonstrated that a weakened AMOC can produce anomalous warming in the Indo-Pacific ocean^{55,56} (Extended Data Fig. 9). The records show that Indian Ocean SST warming relative to the North Atlantic is more attenuated in T-I than T-II, it may suggest that

the AMOC is more active during T-I than T-II (Extended Data Fig. 10). Also, carbon isotope ratios of benthic foraminifera in North Atlantic (ODP Site 983) suggest a more active AMOC strength during T-I compared with the previous four terminations (Extended Data Fig. 11). These observations strengthen the crucial role of AMOC change in determining the onset of permafrost thawing during deglaciation. Therefore, the evident phase difference in onset timing of permafrost thawing between circumarctic and TP (Methods) can be well explained by changes in the strength of the AMOC that exerts opposing thermal effects in the two regions (Fig. 4)."

Smaller comments:

83 "the" is unnecessary before "cave carbonate deposits"

Re: Revised

182 Change "Less clouds increase" to "Less cloud cover increases"

Re: Revised

201-2 Can the authors briefly describe the data and model results cited in this sentence, to help the reader understand what these conclusions are based on?

Re: Revised. See **lines 267-269**.

"This is consistent with previous circumarctic permafrost studies of marine sediments and model simulations suggesting a remobilization and release of permafrost carbon during last deglaciation."

205 Change to "warming effect of reduced cloud cover"

Re: Revised

284 Here and in the data tables and figure captions, I believe " 2δ " should be changed to " 2σ ".

Re: Revised.

634 Why is the focus on boreal summer? Shouldn't the caves be responding to mean annual temperature?

Re: Summer temperatures are presented here because permafrost thawing and the thickness of the active layer reaches its maximum primarily in the summer, when the weakening of the South Asian monsoon due to the AMOC disruption leads to a significant summer warming through local cloud – precipitation – soil moisture feedbacks, and hence resulting in higher mean annual temperatures (Fig. R17), allowing stalagmites to grow. Moreover, the model results also show a warming in mean annual temperatures of TP. We have included the mean annual temperature results into to the discussion, please see **lines 251-253**.

“Collectively, AMOC collapse would significantly contribute to TP summer (Fig. 3a) as well as annual warming (Extended Data Fig. 7) through local cloud – precipitation - soil moisture feedbacks.”

Figure R17 | Tibetan Plateau annual mean temperature anomaly in response to AMOC collapse.

651 This figure should also show the locations of the Canadian and Siberian caves.

Re: Revised.

658 I'm not sure what the δD record (f) is showing. Also, if North Indian Ocean temperatures are important, why aren't they shown for the most recent deglaciation?

Re: The leaf wax δD records from northern Bay of Bengal indicates the precipitation amount changes during recent two interglacial periods. This extended data figure was used to support the hypothesis that the TP stalagmites growth was regulated by temperature rather than precipitation amount, by comparing TP growth phase with temperature and hydrological proxies. Given the less constrained age model of marine records, we removed this extended data figure. Based on the $\delta^{18}O$ records of Tianmen Cave, it is unambiguously that the TP growth start at the Weak Monsoon Interval (WMI) with low rainfall amount, thus ruling out the possibility that the onset of growth was caused by increased water amount (Extended Data Fig. 3).

687 "Benthic" is misspelled in the axis labels of multiple figures. Also, again – why isn't the Chinese speleothem $\delta^{18}O$ record shown here, given that the authors' hypothesis links WMIs to local thaw?

Re: Revised. Chinese speleothem $\delta^{18}O$ records was added. Please see Extended Data Fig. 2 and 3.

REVIEWER COMMENTS

Reviewer #1 (Remarks to the Author):

Review of the manuscript “Asynchronicity of deglacial permafrost thawing controlled by millennial-scale climate variability” by Yan et al.

This is a manuscript reconstructing permafrost history of Tibetan Plateau by dating of 0.5 My speleothem record from Tianmen cave system by U-series chronology. The authors find that unlike Circum-Arctic permafrost, TP permafrost initiates thawing early in deglaciations, coinciding with weak monsoon intervals (WMI) and sluggish Atlantic Meridional Overturning Circulation (AMOC) during Heinrich stadials. The authors write that weakening of South Asian Summer Monsoon induces anomalous TP warming through local cloud – precipitation – soil moisture feedbacks. This, combined with cooling in circum-Arctic regions, results in asynchronous permafrost thawing across the Northern Hemisphere. The authors also argue that during the last deglaciation, however, the early recovery of the AMOC during the Bølling-Allerød, coupled with a persistent AMOC during Heinrich Stadial 1, effectively suppressed the local feedbacks, advancing circum-Arctic permafrost thawing while delaying thawing of permafrost in the TP. Authors argue that basing on their study, permafrost carbon release, influenced by millennial-scale AMOC variability, is a non-trivial contributor to deglacial atmospheric CO₂ rise.

I am generally satisfied with authors reply to my comments in the first review. However, I suggest some corrections to the paragraph (lines 117-133) they modified to address these comments.

I suggest to formulate the first two sentences of the above paragraph as following:

“The cave is located on the plateau few tens of meters above a river. The cave receives its water from direct rain and snowmelt above it. Absence of stream channels above the cave prevents formation of “Taliks” (“holes” or “windows” in the permafrost under the stream channel) through which the water can infiltrate down into the cave.”

RI: We acknowledge the reviewer’s comment. We have complied with the reviewer’s suggestion. Please see lines 347-349.

“The cave is located on the plateau few tens of meters above a river. Absence of stream channels above the cave prevents formation of “Taliks” (unfrozen patches in the permafrost beneath a stream channel) through which the water can infiltrate down into the cave.”

I also don’t agree to the following statement:

“Our results show that speleothem growth in Tianmen cave generally commences within the dry and weak monsoon intervals (WMI) associated with a collapsed AMOC state during terminations³¹ (Fig. 2; Extended Data Fig. 2 and 3). This suggests that the growth of speleothem in the study region is not influenced by the availability of meteoric water and runoff, but rather the thawing water from the overlying permafrost caused by the AMOC collapse-induced warming.”.

The amount of annual precipitation in the study area is 440 mm, which is much higher than any threshold of speleothem precipitation (200-350 mm, (Vaks et al. 2010)). Therefore it is wrong to argue that the permafrost thaw is the only source of water for the speleothem growth. I say that both atmospheric precipitation and permafrost thaw caused growth of speleothems in the study region.

R2: We apologise for the misleading statement here. We have revised and modified the sentence accordingly:

Please see lines 114-119; 125-131; 275-276.

“Speleothem growth in permafrost regions largely depends on a steady drip of water into the cave, which is controlled by the soil's percolation rate associated with the depth of the permafrost active layer. Rainfall and meltwater from ground ice, the main sources of drip water²⁶, can seep into the cave through the overlying soil layers only when the active layer deepens due to permafrost thawing. This facilitates speleothem formation”

“Our findings indicate that speleothem growth in the Tianmen cave typically begins during weak monsoon intervals (WMI) associated with a collapsed AMOC state during climate terminations³² (Fig. 2; Extended Data Fig. 2 and 3). These periods are characterized by drier conditions compared to interglacial climates³³. This suggests that the overlying permafrost conditions, which influence both meltwater availability and soil percolation rates, may be the primary drivers of speleothem growth during terminations, rather than meteoric water, the modern source of drip water to the cave.”

“As a result, associated meltwater as well as precipitation would penetrate through the cave causing speleothem growth.”

Reference:

Vaks, A., et al. (2010). "Middle-Late Quaternary paleoclimate of northern margins of the Saharan-Arabian Desert: reconstruction from speleothems of Negev Desert, Israel." *Quaternary Science Reviews* 29(19-20): 2647-2662.

Reviewer #2 (Remarks to the Author):

The authors have made a substantial effort in revisions, and many of the revisions significantly improve the manuscript (while a few others do not)

There were three main issues in the original manuscript which needed to be addressed in the revision

1) More robust data treatment especially better criteria when some age data are excluded, an aspect significantly affecting the interpretations. This is overall much better resolved in the revision,

because now an analysis is provided which includes all of the available dates and Extended Figure 2 shows this clearly. On the other hand, the new definition of “continuous growth phases “which the authors use to neglect the significance of an early Arctic growth phase at MIS 9, still needs better attention. The criteria of continuous being defined by “overlapping error bars on successive dates” is not an ideal criteria because it depends on the error on the dates (depends on age and technique) and depends on the density of sampling for ages (a growth phase may be continuous but sampled at widely spaced intervals so that age uncertainty may not overlap). The current definition appears chosen ad hoc to allow the authors to disregard this early growth phase. It may be better to acknowledge that it is difficult in slow growing stalagmites to verify if growth is continuous but to indicate that for this study a growth phase is defined as an increment of speleothem deposition supported by multiple consecutive ages without evidence for visual discontinuity of growth. Then acknowledge the long Siberian growth phase preceding MIS 9e and address it in Figure 2 and discuss it.

R1: We thank the reviewer for pointing out this issue. In the new definition, we deleted the criterion of ‘continuous’ defined by “overlapping error bars on successive dates”, which is influenced by sampling density and dating uncertainties. We have complied with the reviewer’s comment and revised the criterion. Accordingly, we revised Figure 2 to highlight the abnormal Siberian growth phase which preceded MIS 9e and added some discussion about it. Please see lines 419-423;170-180.

“For slow-growing speleothem, verifying if the growth is continuous is difficult. In this study, the “continuous growth” phase is defined as an increment of speleothem deposition supported by multiple consecutive ages without evidence for visual discontinuity of growth, and with the primary growth period occurring during warm interglacial periods, as suggested by the established causality between speleothem growth and overlying ground thawing (Extended Data Fig. 2). ”

“During T-II and T-III, the onset of circumarctic permafrost thawing is confined to interglacial periods, and lags TP permafrost thawing (Fig. 2). During T-IV, Siberian speleothems exhibit an unusual growth phase preceding MIS 9e. This is identified through four ages from four individual speleothems, and is better characterized as a 'growth event' rather than a phase of continuous growth. The cause of this glacial growth event is undetermined. Glacial low biogenic silica content in nearby Lake Baikal suggests it is unlikely to have been driven by climate warming (Extended Data Fig. 4). Consequently, this glacial growth event may not be directly linked to continuous permafrost thawing suggesting that the thawing of TP permafrost likely preceded that of the circumarctic region. Given the complexity of thawing mechanisms and the scarcity of regional temperature records, further research is necessary to fully understand the causes of this early growth event.”

Extended data Figure 2 is a very clear and transparent presentation of the data. Figure 2 shows all the dates for Tibet but does not show all the dates for Siberia and Canada, a choice that obscures the complexity of the data. Why not show all the dates for all regions for a transparent presentation of the results in Figure 2 (this would be more important than showing 2 f and 2g which give context but are redundant compared to information in 2e).

R2: Thank you for your helpful suggestions. In the previous version, the Siberian speleothem growth

preceding MIS 9e was not shown because the inaccurate definition of continuous growth excluded this growth event. In the new version, we have revised the definition as well as Figure 2 accordingly, and all dates are presented.

2) Better attention to the chronological uncertainties, and the suggestion that the EASM speleothem records be used for comparison since there is strong evidence they record AMOC

Reviewers 2 and 3 both suggested that the authors base their interpretation of timing on comparison with the absolute dated d18O record of east Asian monsoon speleothems, which is documented to record the impact of AMOC on the Asian monsoon system with periods of weak AMOC manifest in millennial positive anomalies traditionally referred to as “weak monsoon intervals”. The authors have incorporated the EASM d18O record as suggested as well as the Tibetan cave d18O with a citation to its link to the south Asian monsoon. It may be worth emphasizing in the text that these d18O records indicate broad changes in the monsoon circulation during AMOC weakenings, but not necessarily changes in the amount of precipitation at the site. (otherwise stating that something is a “summer monsoon proxy” will lead most readers to conclude the d18O is related to the amount of summer monsoon precipitation, yet the speleothem d18O does not have a consistent relationship with precipitation amount across the Asian monsoon domain). Furthermore statements such as “It has been suggested that reductions in Asian Summer Monsoon during Terminations are a consequence of AMOC slow down during terminal stadials” could also be misleading. Rather, than using the word “reduction” in Asian summer monsoon, indicate “Changes in the Asian summer monsoon circulation manifest in positive d18O speleothem anomalies, are a consequence of AMOC slow down during terminal stadials”.

R3: We thank the reviewer for identifying this issue. We have underlined that changes in ASM speleothem $\delta^{18}\text{O}$ records indicate the large-scale changes in monsoon circulation associated with AMOC disruptions, rather than a simple rainfall proxy. We complied and rephrased the paragraph accordingly. Please see lines 193-198.

“The $\delta^{18}\text{O}$ values of speleothems in the Asian summer monsoon (ASM) domain are primarily controlled by summer monsoon intensity which is driven by large-scale changes in atmospheric circulation accompanied by changes in rainfall amount, temperature, seasonality, as well as moisture source and trajectories (ref. ²⁵ and references therein). It has been suggested that changes in ASM circulation manifest as positive $\delta^{18}\text{O}$ speleothem anomalies, are a consequence of AMOC slow down during terminal stadials^{32,35}.”

Lines 181-188, added in revision, add some not useful clauses and some misinterpretation. First, D18O speleothem signals record different facets of the climate system in different regions, so d18O carbonate from the Arctic is not necessarily an AMOC indicator (especially in the Canadian Arctic). This first sentence is therefore not coherent with the line of argument and can be deleted.

R4: We fully agree with review’s comment. We have deleted the offending sentence.

Most importantly the records cited in this paragraph only cover Termination II. This adds some detail to the results already inferred from the east Asian monsoon record d18O chronology but it does not strengthen the interpretation of Tibet with the weak AMOC state in any other terminations

and the sentences at the end of the paragraph overstate the importance. The citation of TII records can be added as detail adding a further support in the previous paragraph, emphasizing that they provide confirmation for TII supporting the timing inferred from East Asian monsoon records.

R5: Thanks for the comment. We rephrased the paragraph accordingly, added this T-II case at the end of paragraph as a complement. T-II is a well-investigated deglaciation, with well-constrained records, providing a good case supporting our hypothesis. Please see lines 200-208.

“Moreover, the $\delta^{18}O_{\text{carbonate}}$ record of a central European Cave³⁶ and $\delta^{13}C_{\text{carbonate}}$ record from a cave fronting the North Atlantic³⁷ provide good constraints on the timing of AMOC disruption and the last warming stage of the T-II (Extended Data Fig. 5). By comparing our onset dates of continuous growth with these absolute dated record from the high-latitude regions, we find that TP permafrost thawing occurred during AMOC collapse of T-II, while the circumarctic permafrost thawed during the following warm interglacial period (MIS 5e). This well-constrained termination corroborates AMOC-induced TP permafrost thawing inferred from ASM $\delta^{18}O$ records.”

Critically, in this section and the extended figure the authors have misinterpreted the North Atlantic (Iberian) speleothem isotope record. The d18O in NW Iberia is not an AMOC indicator, the d13C from this record is a temperature indicator which is sensitive to the stadial temperature decreases from AMOC weakening, as described in the publication. Therefore the d13C, rather than d18O, is more relevant for this NW Iberia record in the comparison figure and the text need to be corrected so that it does not allude to the d18O of the North Atlantic cave.

R6: We acknowledge the reviewer’s comment. The speleothem $\delta^{18}O$ records of NW Iberia have been replaced with $\delta^{13}C$. We have modified the text and figure accordingly.

Lines 189-200 do not really address the fundamental concern that the marine records do not provide suitable chronology to add any support to the argument. The first sentence is misleading “In contrast to circumarctic permafrost, the TP permafrost thawing generally coincided with deglacial Asian WMI (i.e. AMOC disruptions) during T-II to T-V. This is further supported by the AMOC proxies derived from marine records”

R7: We agree with reviewer’s comment. Less-constrained AMOC proxies derived from marine records can only be treated as a context. We deleted this sentence.

Comparison with marine records cannot be used to support that TP thawing coincided with AMOC disruptions. Rather, the marine records give a richer context for the phasing of the AMOC disruptions relative to the overall progression of terminations and can be used in this way. It is good that the authors have now described the age models for the marine cores and ice core which is useful. Illustrating the IRD and d13C in Figure 2 f and g is OK to give the context if it does not come at the cost of other more vital information. But nothing new is gained by adding new text about the relationship with the timing of IRD peaks. IRD records come from marine sediment cores which have no absolute chronology. Even when the chronology of the marine record includes a few tuning points to the EASM speleothem d18O record, from the point of view of chronology and timing, nothing more is gained for this study by comparing the phases with an IRD record – it is equivalent to comparing directly with the “weak monsoon events” in the EASM record which is why reviewers 2 and 3 both suggested using the EASM record. This paragraph should be adjusted to emphasize

not the chronology and phasing but to enrich the context of the events and highlight that there is a sequence of events and cite the literature which has led to the modern framework for understanding terminations. Something to the effect that:

“Although the chronology for marine records is much less precise than speleothem records, marine records confirm that glacial terminations feature a sequence of changes with increased iceberg rafting and benthic foram $\delta^{13}\text{C}$ evidence for a shallow or weak AMOC at the onset of the termination preceding the main deglaciation as manifest in benthic $\delta^{18}\text{O}$. This has been widely interpreted as early deglacial freshwater addition disrupting AMOC”

R8: We acknowledge the reviewer’s comment. We have rewritten the paragraph accordingly. Please see lines 209-217.

“While the chronology of marine records is less precise than that of absolutely dated speleothems, various lines of marine evidence consistently demonstrate that glacial terminations are characterized by a sequence of prominent climatic events. These include increased iceberg rafting (Fig. 2g) and depleted benthic foraminifera $\delta^{13}\text{C}$ values in the North Atlantic (Fig. 2f), indicating a shallower or weaker AMOC at the onset of the termination³⁸, which precedes the main phase of a deglaciation as manifested in benthic $\delta^{18}\text{O}$ records (Fig. 2). This has been widely interpreted as early deglacial freshwater input disrupting AMOC³⁸. These “Heinrich Events” are well reflected in absolutely dated ASM speleothem records and support the link between continuous TP permafrost thawing and early deglacial AMOC disruptions.”

3) A key recommendation was more nuance in the interpretation and better attention to the variation in phasing where it is not always possible to document that Tibet leads Arctic thawing given the dating uncertainties.

This needs to be better acknowledged in the paragraph from lines 167 to 180, and lines 210-212 can highlight this better. Cutting unnecessary text in previous sections as suggested in comment 2, will give room to better acknowledge this. The dataset is good and honesty about it will only strengthen the paper and lead to more effective future modeling and data collection. Acknowledge in which terminations it is clear that Tibet leads Arctic responses. (line 170 to 171 say Tibet onset is “generally within deglaciations” with the exception of TI, but this “generalization” should be followed by a statement that clearly indicates in which cases the onset is well constrained to be in the deglaciation, in which cases the onset is likely (within error) of the deglacial stadial/WMI, and in which cases the onset is not (e.g. TI)

R9: We fully agree with the reviewer’s comment. The large dating uncertainties, particularly in the circumarctic samples, make it challenging to establish a clear lead-lag relationship between the two regions in some terminations. We have acknowledged this in the section and revised the text accordingly. We included a detailed discussion to emphasize the nuanced differences in each termination, particularly T-IV and T-V, where the dating uncertainties are relatively large and an anomalous early growth event was observed.

Large dating uncertainties, especially in those circumarctic samples, making it difficult to document clear lead-lag relationship between two regions in some cases. We acknowledged this in the section and rephased the text, a detailed discussion was added to highlight the nuanced differences in each termination, particularly T-IV and T-V, in which the dating uncertainties is relatively large and an anomalous early growth event was observed. Please see lines 170-188; 244-247.

“During T-II and T-III, the onset of circumarctic permafrost thawing is confined to interglacial periods, and lags TP permafrost thawing (Fig. 2). During T-IV, Siberian speleothems exhibit an unusual growth phase preceding MIS 9e. This is identified through four ages from four individual speleothems, and is better characterized as a 'growth event' rather than a phase of continuous growth. The cause of this glacial growth event is undetermined. Glacial low biogenic silica content in nearby Lake Baikal suggests it is unlikely to have been driven by climate warming (Extended Data Fig. 4). Consequently, this glacial growth event may not be directly linked to continuous permafrost thawing suggesting that the thawing of TP permafrost likely preceded that of the circumarctic region. Given the complexity of thawing mechanisms and the scarcity of regional temperature records, further research is necessary to fully understand the causes of this early growth event. During T-V, the large dating uncertainties in circumarctic samples make it challenging to establish the lead-lag relationship between TP and circumarctic permafrost thawing. Nevertheless, when considering the dates with relatively lower age uncertainties as the onset of continuous speleothem growth (Methods), it is possible that Tibetan Plateau permafrost thawing may still have preceded that of the circumarctic region. Overall, despite the complexities in the timing of circumarctic permafrost thawing, the onset of Tibetan Plateau permafrost thawing generally occurs within dating uncertainties during deglaciations, prior to the continuous thawing of circumarctic permafrost during warm interglacial periods (except for T-I) (Fig. 2 and Extended Fig. 2).”

“Consequently, the lead-lag relationships between Tibetan Plateau and circumarctic permafrost thawing may vary among different terminations. To clarify these differences, more precisely dated speleothems from permafrost regions covering T-IV and earlier periods are needed.”

The addition of analysis of temperature profiles and permafrost thawing from the model runs was an interesting addition.

R10: We thank the reviewer’s recognition.

Reviewer 3 indicated that the case for permafrost CO₂ release was overly substantiated but this is still the first sentence of the abstract, so it may be necessary to revise the abstract.

R11: We rephased the abstract accordingly. Please see lines 30-31; 25-46.

*“Permafrost carbon release is **important** in explaining deglacial CO₂ changes alongside deep-sea carbon outgassing.”*

*“Our results indicate that permafrost carbon release, influenced by millennial-scale AMOC variability, **may have been** a non-trivial contributor to deglacial atmospheric CO₂ rise.”*

The revised manuscript contains issues in the English language that should be addressed before the manuscript can be considered for publication. Contractions are not used in formal English, other spelling and grammatical errors should be fixed. Usage such as “ We assume this is related to regional change” is not really right for a sentence forwarding a hypothesis tested with a model, rather say r “ One possible cause is regional change on the Tibetan plateau...” or “ our favored explanation is “. Also usage such as “ We want to highlight that.” is not stylistically used. Better,

“ It is noteworthy that... “ The author list includes coauthors whose first language is English should be able to address these and other issues if they take time to read the manuscript carefully.

R12: We thank the reviewer for pointing out the writing issues in such detail. The language in our manuscript has been revised by native English-speaking co-authors. We have carefully reviewed and revised the entire manuscript to address the English language issues. This includes eliminating contractions and correcting spelling and grammatical errors.

Reviewer #3 (Remarks to the Author):

I appreciate the revisions the authors have made, and in particular the addition of the TM d18O records in Supp. Fig 3 and the heat diffusion modeling. I have three further requests for revisions.

R1: We thank reviewer’s recognition.

First, the last sentence of the abstract is too strong. This sentence states that the authors show that permafrost carbon release “is a non-trivial contributor to deglacial atmospheric CO₂ rise.” The authors show that permafrost carbon may have contributed to deglacial atmospheric CO₂ rise, but they do not show that it is "non-trivial", as they do not provide constraints on how much carbon was released from permafrost. I suggest changing to "may have been a non-trivial contributor to deglacial atmospheric CO₂ rise."

R2: We fully agree reviewer’s comment. We revised the abstract accordingly. Please see lines 30-31; 25-46.

*“Permafrost carbon release is **important** in explaining deglacial CO₂ changes alongside deep-sea carbon outgassing.”*

*“Our results indicate that permafrost carbon release, influenced by millennial-scale AMOC variability, **may have been** a non-trivial contributor to deglacial atmospheric CO₂ rise.”*

Second, the new text added to the manuscript has multiple small grammatical errors that should be fixed prior to publication. Review of this new text by all co-authors should be sufficient to address these errors.

R3: We thank reviewer’s comment. We have carefully reviewed and revised the entire manuscript to address the English language issues. The language in our manuscript has been revised by native English-speaking co-authors.

Third, the first section of the methods needs more information about the cave. How deep is the cave? Is it relatively horizontal, or does it slant downwards/upwards? Approximately how thick is the epikarst above the cave?

R4: We thank reviewer’s comment. We have added more available information about Tianmen cave system. Please see lines 349-350.

“The cave is less than 50 meters deep and extends relatively horizontally. The epikarst above the cave is approximately 10 meters thick on average.”

REVIEWERS' COMMENTS

Reviewer #1 (Remarks to the Author):

I see that the last sections that needed correction were modified. Therefore, I think this manuscript is ready for publication.

Re: Thank you for your thoughtful comments and suggestions throughout the review process.

Reviewer #3 (Remarks to the Author):

I am satisfied by the authors' revisions, with the exception that line 30 should be changed to, "Permafrost is a potentially important source of deglacial carbon release alongside..."

Re: We have made the change as recommended to. Thanks again for your constructive feedback and support, which largely improved the quality of the manuscript.